# Multi-step control of homologous recombination via Mec1/ATR suppresses chromosomal rearrangements

Bokun Xie [1], Ethan James Sanford [1], Shih-Hsun Hung[2], Mateusz Wagner [1], Wolf-Dietrich Heyer[2] & Marcus B Smolka [1]✉

## Abstract

**The Mec1/ATR kinase is crucial for genome stability, yet the mechanism by which it prevents gross chromosomal rearrangements (GCRs) remains unknown. Here we find that in cells with deficient Mec1 signaling, GCRs accumulate due to the deregulation of multiple steps in homologous recombination (HR). Mec1 primarily suppresses GCRs through its role in activating the canonical checkpoint kinase Rad53, which ensures the proper control of DNA end resection. Upon loss of Rad53 signaling and resection control, Mec1 becomes hyperactivated and triggers a salvage pathway in which the Sgs1 helicase is recruited to sites of DNA lesions via the 911-Dpb11 scaffolds and phosphorylated by Mec1 to favor heteroduplex rejection and limit HR-driven GCR accumulation. Fusing an ssDNA recognition domain to Sgs1 bypasses the requirement of Mec1 signaling for GCR suppression and nearly eliminates D-loop formation, thus preventing non-allelic recombination events. We propose that Mec1 regulates multiple steps of HR to prevent GCRs while ensuring balanced HR usage when needed for promoting tolerance to replication stress.**

**Keywords** Mec1; Sgs1; Resection; Homologous Recombination; Chromosomal Rearrangement
**Subject Category** DNA Replication, Recombination & Repair

## Introduction

Gross chromosomal rearrangements (GCRs) are aberrant structural variations in chromosomes, such as deletions, translocations and amplifications that compromise genomic stability and drive oncogenesis (Mitelman et al, 2007; Gordon et al, 2012; Cancer Genome Atlas Research Network et al, 2013). An important source of GCRs is DNA replication stress. The progression of replication forks is often impeded by various types of barriers, such as DNA lesions, difficult-to-replicate regions, and transcriptional intermediates, leading to stalled replication fork structures that can be converted into double-strand breaks (DSBs) through the action of nucleases (Lambert et al, 2005; Lemoine et al, 2005; Mizuno et al, 2009; Aksenova et al, 2013; Helmrich et al, 2013; García-Muse and Aguilera, 2016). During DNA replication, DSBs are commonly repaired via homologous recombination (HR), a multi-step process that includes DNA end resection, strand invasion, DNA synthesis and the processing of recombination intermediates (Mimitou and Symington, 2009; Sanchez et al, 2021; Heyer, 2015). HR is a high-fidelity mode of DNA repair, helping to prevent genomic rearrangement and maintain the overall integrity of the genome when sister chromatids are used as templates. However, when strand invasion occurs at the wrong locus, non-allelic HR between partially homologous (homeologous) sequences can happen, leading to the formation of GCRs (Putnam and Kolodner, 2017; Al-Zain and Symington, 2021). This includes the formation of heteroduplex DNA between the homeologous sequences which is subject to recognition by the mismatch repair system and heteroduplex rejection to avert GCRs (Spies and Fishel, 2015). How cells regulate HR-mediated DNA repair to prevent non-allelic recombination and GCRs is not fully understood. In particular, how cells balance the use of HR to ensure its adequate use while discerning from contexts where it may drive GCR events is a complex problem that likely requires decision-making steps and sensing mechanisms.

Understanding the mechanisms of GCR suppression in higher eukaryotes is challenged by the lack of sensitive and effective assays for monitoring GCRs that can be coupled to genetic screens. In contrast, significant progress in the study of the genesis of GCRs has been made using the "classical" GCR assay based on canavanine and 5-fluoroorotic acid (5-FOA) selection in *S. cerevisiae* to screen for spontaneous GCRs associated with the combined loss of the *CAN1* and *URA3* genes placed at the non-essential left arm of chromosome V (Chen and Kolodner, 1999). Using this approach, numerous factors implicated in DNA repair and DNA damage checkpoint have been identified to play pivotal roles in suppressing GCRs, among them the Mec1/ATR and the Tel1/ATM kinases (Myung et al, 2001a, 2001b, 2001c; Putnam and Kolodner, 2017). While deletion of *MEC1* leads to significant increases in GCR rates (~200-fold higher compared to WT) and deletion of *TEL1* has no

[1]Department of Molecular Biology and Genetics, Weill Institute for Cell and Molecular Biology, Cornell University, Ithaca, NY, USA. [2]Department of Microbiology and Molecular Genetics, University of California, Davis, Davis, CA, USA. ✉E-mail: mbs266@cornell.edu

effects on GCR rates, $mec1\Delta$ $tel1\Delta$ cells display one of the highest GCR rates reported (over 10,000-fold increase compared to WT) (Myung et al, 2001c). Despite the crucial roles of Mec1 and Tel1 in GCR suppression, the mechanism by which these kinases prevent GCR accumulation remains incompletely understood.

Mec1 is a phosphoinositol-3-Kinase-like kinase (PIKK) that functions as a sensor of DNA replication stress by recognizing single-strand DNA (ssDNA) accumulation mainly at stalled replication forks and recessed DSBs (Zou and Elledge, 2003; Matsuoka et al, 2007; Smolka et al, 2007; Bastos de Oliveira et al, 2015). Mec1 recognizes replication protein A (RPA)-coated ssDNA via its cofactor Ddc2 (Zou and Elledge, 2003; Deshpande et al, 2017) and, once recruited, is activated by proteins such as Dpb11, Ddc1 and Dna2 that contain a disordered Mec1-activating domain (Mordes et al, 2008; Navadgi-Patil and Burgers, 2008, 2009; Kumar and Burgers, 2013). Active Mec1 phosphorylates and activates the downstream kinase Rad53 to initiate the canonical DNA damage checkpoint response that promotes cell cycle arrest, fork stabilization and protection, inhibition of origin firing, regulation of dNTP production, and transcriptional reprogramming (Sanchez et al, 1997; Segurado and Tercero, 2009; Yekezare et al, 2013; Zhao et al, 2001; Bastos de Oliveira et al, 2012). The classical checkpoint adaptor Rad9 contributes to transducing signaling from Mec1 to Rad53, while also playing roles in the control of DNA end resection, the first step in HR-mediated DNA repair (Gilbert et al, 2001; Schwartz et al, 2002; Clerici et al, 2014). Mec1 has also been reported to play roles in the regulation of HR-mediated DNA repair independently of its canonical function in checkpoint signaling (Barlow and Rothstein, 2009; Flott et al, 2011; Ullal et al, 2011; Dion et al, 2012; Bashkirov et al, 2000). Depending on the context, Mec1 can exert inhibitory or stimulatory effects on DNA end resection control. For example, while early in the response to DNA lesions Mec1 can inhibit resection by facilitating the recruitment and oligomerization of the resection antagonist Rad9 at DNA lesions (Clerici et al, 2014; Ferrari et al, 2015), at later stages Mec1 can then promote long-range resection by mediating the recruitment of the DNA repair scaffolding protein Slx4, which counteracts the resection block formed by Rad9, therefore promoting resection (Dibitetto et al, 2016; Liu et al, 2017). The recruitment of both Rad9 and Slx4 relies on their interaction with Dpb11, a multi-BRCT domain scaffold that recognizes phosphorylated Rad9 or Slx4 and stabilizes them at DNA lesions (Pfander and Diffley, 2011; Cussiol et al, 2015). In addition to resection control, Mec1 regulates strand exchange through the phosphorylation of the strand exchange factor Rad55 (Herzberg et al, 2006; Janke et al, 2010) and of the recombinase Rad51 (Flott et al, 2011). Mec1 phosphorylation has been proposed to control the ATPase activity of Rad51 and influence HR (Flott et al, 2011).

The ability of Mec1 to suppress GCRs is largely independent of its canonical function in activating the DNA damage checkpoint (Myung et al, 2001c; Lanz et al, 2018). This is best evidenced by the lower rates of GCRs in cells lacking $RAD53$ compared to the rates observed in cells lacking $MEC1$ (Myung et al, 2001c). Despite strong genetic evidence pointing to a crucial checkpoint-independent role for Mec1 in GCR suppression, the precise mechanism by which Mec1 signaling promotes such suppression remains unknown.

To characterize the checkpoint-independent role of Mec1 in GCR suppression, here we monitored Mec1 signaling in $rad53\Delta$

cells using phosphoproteomics and find that loss of the DNA damage checkpoint triggers hyperactivation of Mec1 signaling and hyperphosphorylation of the Sgs1 helicase, a helicase involved in multiple steps of HR, including resection, heteroduplex rejection, and dissolution (Zhu et al, 2008; Ira et al, 2003; Sugawara et al, 2004; Mankouri et al, 2011). In checkpoint-defective cells, GCRs are largely suppressed by Mec1-dependent recruitment of Sgs1 to sites of DNA lesions via phosphorylation of the 9-1-1 clamp and Sgs1, which assembles a 911-Dpb11-Sgs1 complex that increases heteroduplex rejection. Fusing an ssDNA recognition domain to Sgs1 (RBD-Sgs1 chimera) bypasses the requirement of Mec1 signaling for GCR suppression and results in lower D-loop levels, consistent with a model that Mec1 suppresses GCRs by promoting heteroduplex rejection and HR quality control, thus preventing non-allelic recombination events. We propose that Mec1 prevents GCRs through a redundant system of HR control involving both resection control via checkpoint activation and heteroduplex rejection via Sgs1 recruitment and regulation. GCRs drastically rise in cells lacking $MEC1$ due to the abolishment of both GCR suppressing functions.

## Results

### Loss of *RAD53* or *RAD9* triggers Mec1 hyperactivation and dependency on Sgs1 for GCR suppression

Loss of MEC1 causes a ~200-fold increase in the rates of GCRs, while cells lacking $RAD53$ exhibit only a ~30-fold increase in GCR rates (Myung et al, 2001c). Since the loss of Rad53 impairs fork stabilization and resection control, which are expected to contribute to promoting GCR events, we reasoned that Mec1 must promote GCR suppression in $rad53\Delta$ cells via a Rad53-independent signaling response (Fig. 1A). To test this prediction, we compared the phosphoproteome of wild-type and $rad53\Delta$ cells using quantitative mass spectrometry and searched for Mec1-dependent signaling events triggered by checkpoint deficiency (Fig. 1B; Dataset EV1). Mec1-dependent phosphorylation was determined by crossing the dataset with previously reported phosphoproteomic analyses comparing wild-type to $mec1\Delta$ cells (Bastos de Oliveira et al, 2015; Sanford et al, 2021). As expected, S/T-bulky hydrophobic amino acid ($\psi$) motif, the Rad53 phosphorylation motif, was enriched in the set of phosphorylation events downregulated in $rad53\Delta$ cells (Appendix Fig. S1). In contrast, phosphorylation events upregulated in $rad53\Delta$ cells exhibited a significant enrichment of the S/T–Q motif (Fig. 1B,C), the preferential phosphorylation motif for Mec1 (Smolka et al, 2007), indicating that loss of Rad53 triggers hyperactivation of Mec1 signaling.

We previously reported that loss of Rad9, an adaptor protein that promotes Rad53 activation and the control of DNA end resection (Gilbert et al, 2001; Schwartz et al, 2002; Clerici et al, 2014), triggers hyperactivation of a specialized mode of Mec1 signaling targeting proteins associated with ssDNA transactions, including Sgs1, Rfa2, and Uls1 (Sanford et al, 2021). Interestingly, these proteins were also hyperphosphorylated in cells lacking $RAD53$ (Fig. 1D), suggesting that such a response is triggered by a defect common to cells lacking $RAD9$ or $RAD53$. One important function of Rad53 is to preserve the integrity of replication forks (Tercero and Diffley, 2001; Tercero et al, 2003). Consistent with the fork protection role, $rad53\Delta$ cells display

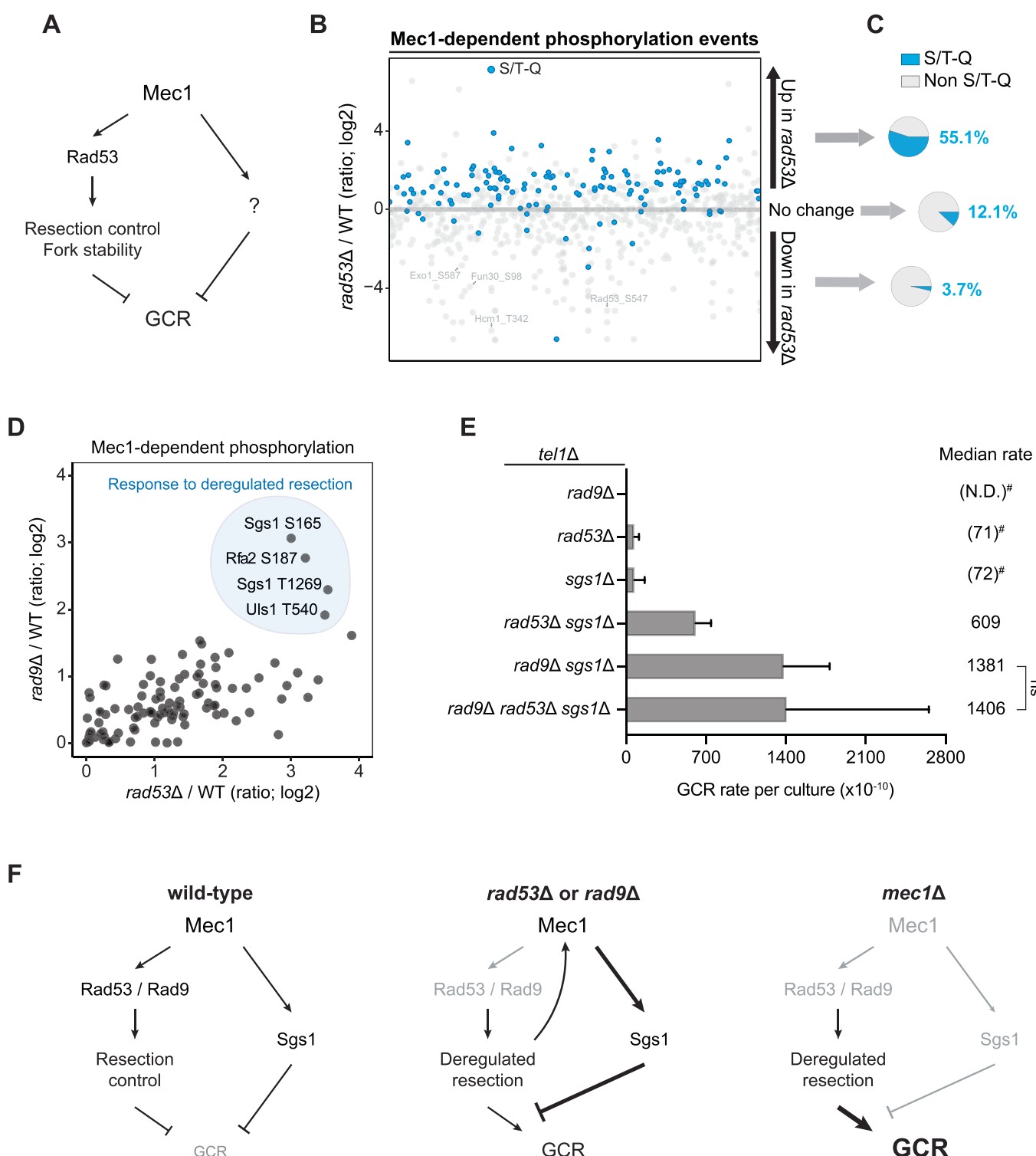

increased Rad52 foci during normal growth and heavily rely on HR for survival (Appendix Fig. S2A,B). In contrast, *rad9Δ* cells do not increase demand for HR (Appendix Fig. S2A,B), as Rad53 can still be active via the Mrc1 adaptor and prevent fork collapse (Alcasabas et al, 2001; Osborn and Elledge, 2003). Thus, we reasoned that signaling hyperactivation is not triggered by replication fork collapse, but most likely due to increased DNA end resection, an outcome observed in both *rad53Δ* and *rad9Δ* cells (Ferrari et al, 2015; Lazzaro et al, 2008; Segurado and Diffley, 2008; Gobbini et al, 2015). Moreover, we hypothesized that the observed hyperphosphorylation of Sgs1, a key helicase involved in multiple steps of HR and GCR suppression (Zhu et al, 2008; Ira et al, 2003; Sugawara

Figure 1. The absence of Rad53 or Rad9 induces Mec1 hyperactivation and a reliance on Sgs1 for GCR suppression.

(A) Proposed model for Mec1-dependent pathways involved in GCR suppression. (B) Quantitative phosphoproteomic dataset showing the modulation of Mec1-dependent phosphorylation events in cells lacking *RAD53*, with S/T–Q consensus motifs (preferential Mec1 phosphorylation sites) indicated in blue. Cells were treated with 0.02% MMS for 2 h. The complete list is available in Dataset EV1. (C) Pie charts showing an enrichment for S/T–Q consensus in the set of phosphorylation events upregulated in *rad53Δ* cells. (D) Quantitative phosphoproteomic data showing Mec1-dependent phosphorylation events upregulated in both *rad9Δ* and *rad53Δ* cells. Among the most highly upregulated sites are residues in Sgs1, Uls1, and Rfa2. (E) Measurement of GCR rates in cells with the indicated genotypes. Bars represent median values and error bars represent standard deviation from 32 individual colonies. "N.D." indicates "not detected". *P* value was calculated using a two-tailed, unpaired *t* test. (#) The approach used for measuring GCRs is unable to accurately measure GCR rates below $500 \times 10^{-10}$. (F) Proposed model for the involvement of Sgs1 in Mec1-dependent GCR suppression. Source data are available online for this figure.

et al, 2004; Mankouri et al, 2011; Putnam et al, 2009; Piazza et al, 2019), evokes a salvage pathway that suppresses GCRs in *rad53Δ* and *rad9Δ* cells. Consistent with this model, the deletion of *SGS1* displayed synergistic effects on GCR rates when combined with the deletion of *RAD53* or *RAD9* (Fig. 1E). The assay was performed in a strain lacking *TEL1*, since it can partially compensate for the loss of *MEC1* in GCR (Myung et al, 2001c). Taken together, these findings are consistent with a model whereby Mec1 suppresses GCRs through distinct pathways, one involving the control of Rad9 and Rad53, and another through the control of Sgs1 (Fig. 1F). Together with our previous report showing that DNA end hyper-resection triggers Mec1 phosphorylation of Sgs1 (Sanford et al, 2021), our findings also suggest that upon loss of DNA end resection control via Rad53 or Rad9, the Mec1-Sgs1 pathway functions as a salvage response important to limit GCRs. Notably, since the GCR rate of *rad53Δ sgs1Δ* cells is lower than the GCR rate of *mec1Δ* cells (Appendix Fig. S3), GCR suppression via Mec1 signaling likely involves other substrates that remain to be identified.

## Deregulated resection increases the demand for Mec1 control of Sgs1 in GCR suppression

To further substantiate the model proposed in Fig. 1F, we investigated the importance of the phosphorylation of Sgs1 by Mec1 for GCR suppression. Sgs1 contains nine serine or threonine residues located in the preferred motif for Mec1 phosphorylation (S/T–Q sites) (Fig. 2A). We mutated all of these nine serine/threonine residues to alanine, yielding the Sgs1^9mut mutant. Sgs1^9mut did not show notable changes in the protein stability (Appendix Fig. S4A). Whereas expression of Sgs1^9mut did not have a detectable effect on GCR rates in *tel1Δ rad9Δ* cells (Fig. 2B), we noticed that expression of Sgs1^9mut in cells lacking *EXO1*, an exonuclease involved in DNA end resection (Zhu et al, 2008; Mimitou and Symington, 2009), resulted in increased GCR rates. The rates of GCR accumulation caused by the expression of Sgs1^9mut were drastically increased in cells lacking both *EXO1* and *RAD9* (Fig. 2B), further consistent with the notion that deregulation of DNA end resection increases the demand for the Mec1-Sgs1 pathway of GCR suppression. Sgs1^9mut also leads to a detectable increase in GCRs in *rad53Δ exo1Δ* cells (Appendix Fig. S4B).

Recently we reported that Mec1 signaling promotes the interaction of Sgs1 with Dpb11 and, indirectly, to the 911 clamp, to recruit Sgs1 to DNA lesions (Sanford et al, 2021). Consistent with our model proposing that the control of Sgs1 via Mec1 signaling is important for GCR suppression, deletion of the N-terminal acidic patches of Sgs1 (Sgs1^APΔ mutant) that mediate the Dpb11-Sgs1 interaction (Sanford et al, 2021) displayed a strong

increase in GCR rate in cells lacking *RAD9*, *EXO1* and *TEL1* (Fig. 2C). Sgs1^APΔ also failed to effectively inhibit GCRs in *rad53Δ exo1Δ* cells (Appendix Fig. S4C). Importantly, the high rates of GCRs observed in *tel1Δ rad9Δ exo1Δ sgs1Δ* cells expressing Sgs1^APΔ were largely dependent on Rad52 (Fig. 2C), consistent with the model that these GCRs are originating due to deregulated HR.

In addition to promoting the Sgs1-Dpb11 interaction, our previous work proposed that Mec1 also promotes the recruitment of Dpb11-Sgs1 to DNA lesions by phosphorylating the Ddc1 component of the 911 clamp, which is recognized by one of the BRCT domains of Dpb11 (Sanford et al, 2021). We therefore measured GCR rate in cells expressing the T602A mutant of Ddc1 that is not recognized by Dpb11 (Puddu et al, 2008). As expected, expression of Ddc1^T602A increased GCR rates in *tel1Δ rad9Δ exo1Δ* cells (Fig. 2D), consistent with the results obtained with Sgs1^APΔ. Surprisingly, the combination of Sgs1^9mut and Ddc1^T602A showed a synergistic effect on GCR suppression (Fig. 2D), suggesting that Mec1-dependent phosphorylation of Sgs1 has roles other than promoting the recruitment of Sgs1 to 911 clamp (via Dpb11). Collectively, these findings support a model in which the control of Sgs1 by Mec1 prevents GCRs driven by non-allelic HR that accumulate in cells deficient for Rad9 or Rad53 (Fig. 2E). Furthermore, the Mec1-Sgs1 pathway appears to be particularly important when the control of DNA end resection is perturbed such as in the absence of Rad53 or Rad9 and, especially, upon further deletion of *EXO1*. Consistent with this notion, the nuclease activity of Exo1 is important for GCR suppression (Appendix Fig. S5). While we favor the model that deregulated resection increases the demand for Sgs1 control by Mec1, we note that it remains possible that the reason for the increased importance of Sgs1 phosphorylation in the absence of *EXO1* may be related to resection-independent functions of Exo1.

## Engineered Sgs1 recruitment suppresses GCRs in Mec1-deficient cells

Based on our proposed model (Fig. 2E), the role of Mec1 in promoting the recruitment of Sgs1 is crucial for GCR suppression, especially in cells lacking proper regulation of DNA end resection. To further test this model, we fused Sgs1 to an RPA-binding domain (RBD; amino acids 1–72 of Ddc2), with the prediction that the RBD-Sgs1 chimera would bypass the requirement of Mec1 for GCR suppression by directly recruiting Sgs1 to ssDNA at recombination intermediates, increasing heteroduplex rejection (Fig. 3A,B). Expression of RBD-Sgs1 significantly impaired break-induced replication (Fig. 3D), consistent with the expected increase in heteroduplex rejection. We have recently reported a similar

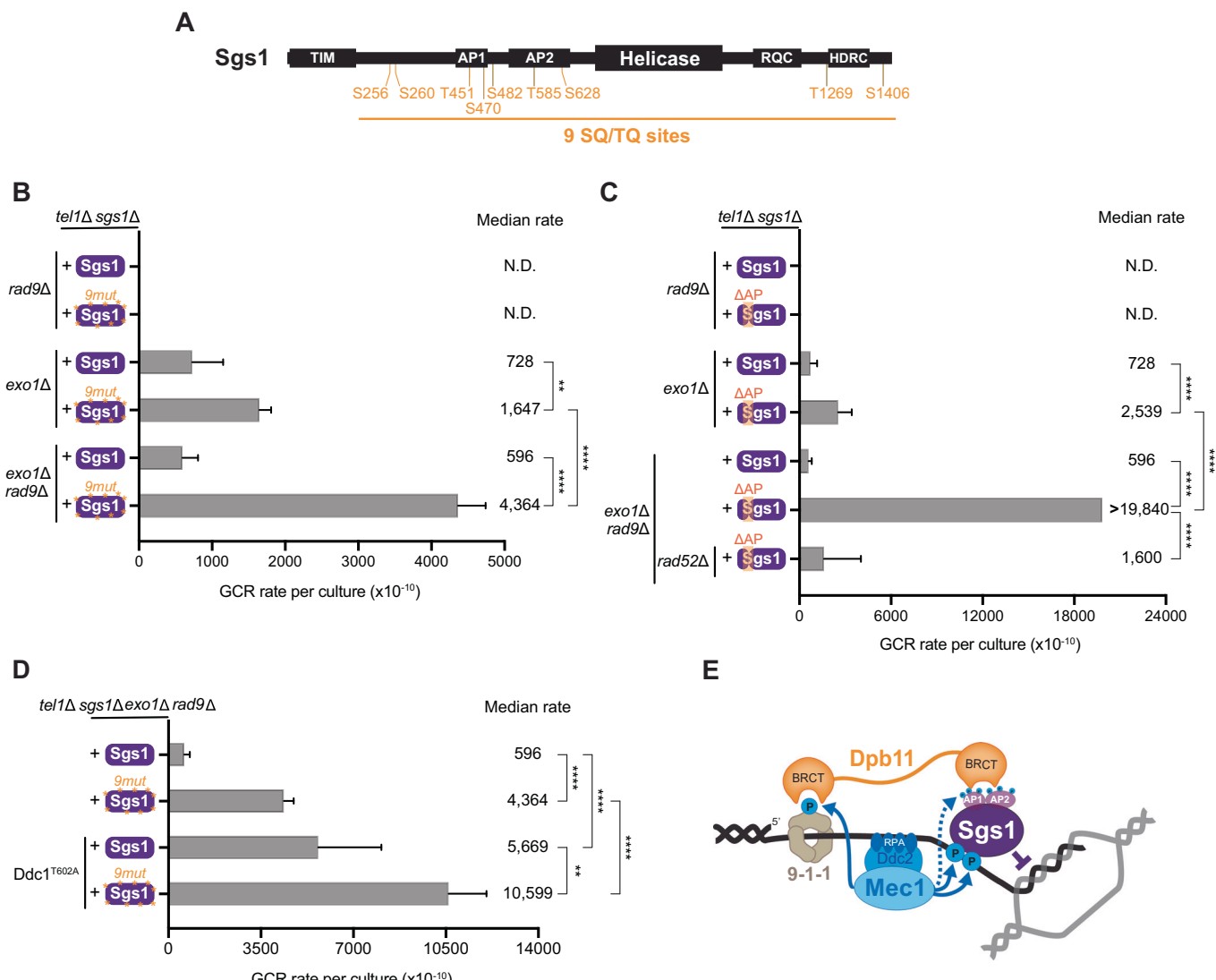

**Figure 2. Deregulated resection increases the requirement for Mec1-dependent phosphorylation of Sgs1 in GCR suppression.**

(A) Schematics of Sgs1 domain architecture indicating the position of SQ/T–Q sites. TIM: Top3 interacting motif; AP1/AP2: acidic patch; RQC: a region found only in the RecQ helicase family (Huber et al, 2006); HDRC: Helicase and RNaseD C-terminal domain. (B) Measurement of GCR rates in cells with the indicated genotypes expressing either Sgs1 or Sgs1$^{9mut}$. Bars represent median values and error bars represent standard deviation from 32 independent colonies. "N.D." indicates "not detected". $P$ value was calculated using a two-tailed, unpaired $t$ test. **$P \leq 0.01$; ****$P \leq 0.0001$. (C) Measurement of GCR rates in cells with the indicated genotypes expressing either Sgs1 or Sgs1$^{AP\Delta}$. Bars represent median values and error bars represent standard deviation from 32 independent colonies. "N.D." indicates "not detected". $P$ value was calculated using a two-tailed, unpaired $t$ test. ****$P \leq 0.0001$. (D) The synergistic effect between Ddc1$^{T602A}$ and Sgs1$^{9mut}$ on GCR suppression. Bars represent median values and error bars represent standard deviation from 32 independent colonies. "N.D." indicates "not detected". $P$ value was calculated using a two-tailed, unpaired $t$ test. **$P \leq 0.01$; ****$P \leq 0.0001$. (E) Speculative model for the mechanism of GCR suppression through Mec1-dependent regulation of Sgs1 that favors heteroduplex rejection. The model is based partly on our previous work showing that Mec1 mediates the recruitment of Sgs1 via the Dpb11 adaptor (Sanford et al, 2021). Source data are available online for this figure.

effect using a fusion between Sgs1 with the BRCT domain 3/4 of Dpb11 (Dpb11$^{BRCT3/4}$-Sgs1) (Sanford et al, 2021), which recruits Sgs1 via recognition of the 911 clamp that is phosphorylated by Mec1, as shown in the model in Fig. 2E. We also reported that expression of Dpb11$^{BRCT3/4}$-Sgs1 causes MMS sensitivity, presumably due to hyper-engagement of Sgs1 preventing HR-mediated DNA repair. Importantly, here we find that RBD-Sgs1 causes MMS sensitivity in both wild-type and *mec1Δ* cells, whereas Dpb11$^{BRCT3/4}$-Sgs1 does not cause MMS sensitivity in *mec1Δ* cells (Fig. 3E). This

finding is consistent with the prediction of hyper-recruitment of Sgs1 via the RBD fusion not requiring Mec1 signaling. Strikingly, RBD-Sgs1 increased Rad52 foci under both normal and MMS-treated conditions (Fig. 3F,G), which could be the consequence of increased DNA damage or a slower repair process. To test whether the expression of RBD-Sgs1 generated increased DNA damage, we monitored the activation of Rad53. Expression of RBD-Sgs1 itself did not elicit Rad53 activation, nor did it impede the regular Rad53 signaling after MMS treatment (Fig. 3H). Collectively, our

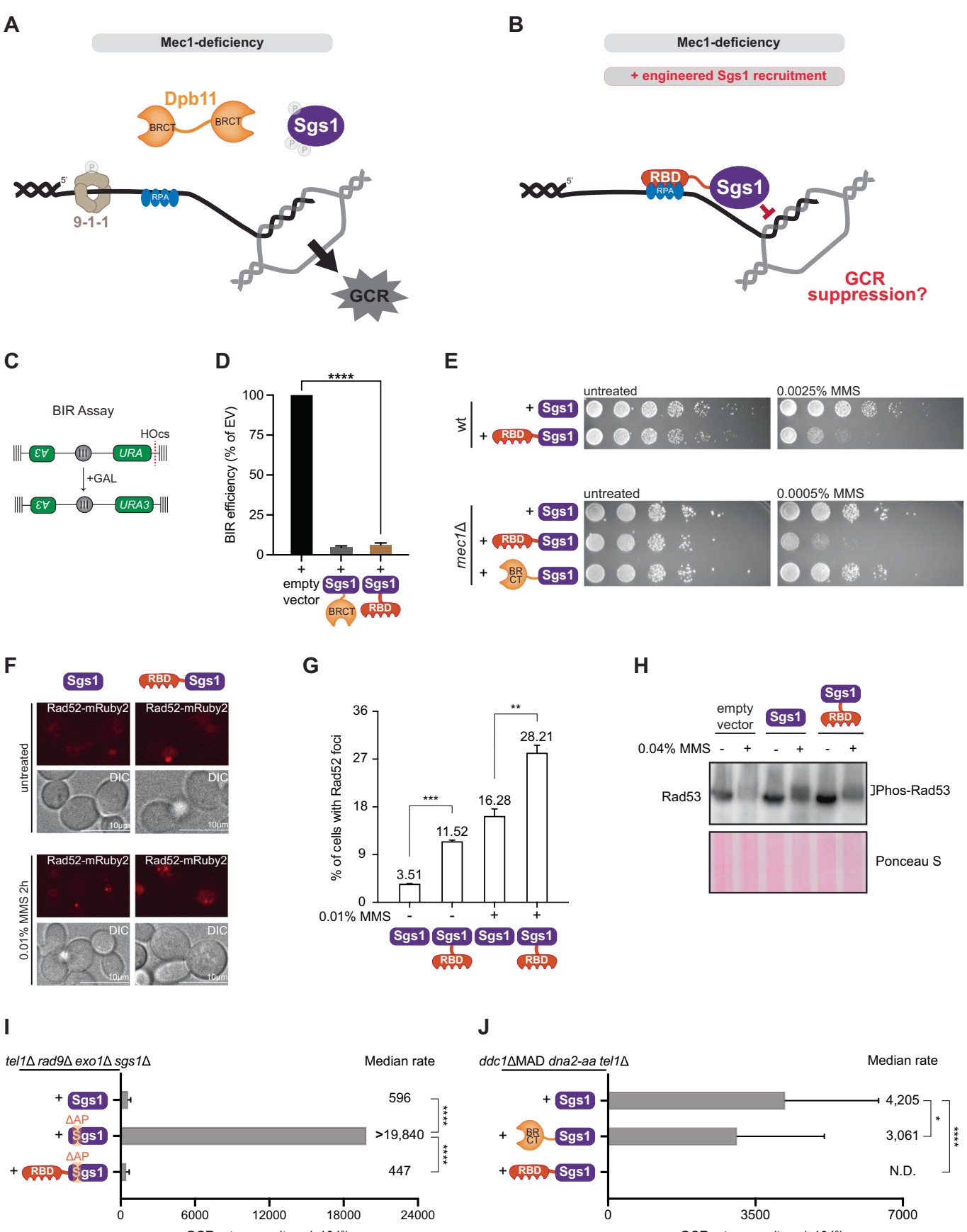

**Figure 3. Engineered Sgs1 recruitment suppresses GCRs in Mec1-deficient cells.**

(A) Schematics illustrating how the lack of Mec1-mediated Sgs1 recruitment leads to increased GCRs. (B) Schematics depicting the rationale for designing an RBD-Sgs1 chimera for recruitment of Sgs1 independently of Mec1 signaling. (C) Diagram of the break-induced replication (BIR) assay used in this study (Sanford et al, 2021). Red line represents a galactose-inducible HO endonuclease cut site. (D) Measurement of BIR efficiency in cells carrying an empty vector or expressing different Sgs1 chimeras. Bars represent mean values and error bars represent standard deviation from three replicate experiments. *P* value was calculated with a two-tailed, unpaired *t* test. ****$P \leq 0.0001$. (E) Dilution assay for monitoring MMS sensitivity of wild-type or *mec1Δ* cells expressing RBD-Sgs1 or Dpb11$^{BRCT3/4}$-Sgs1. (F) Representative image of Rad52 foci in cells expressing Sgs1 or RBD-Sgs1 untreated or treated with 0.01% MMS for 2 h. (G) Quantification of percentages of cells with Rad52 foci from (E). Over 150 cells were scored per replicate. Bars represent mean values and error bars represent the standard error of the mean from three replicate experiments. *P* value was calculated using a two-tailed, unpaired *t* test. **$P \leq 0.01$; ***$P \leq 0.001$. (H) Western blot showing Rad53 mobility shift induced by MMS in cells expressing either Sgs1 or RBD-Sgs1. (I) Measurement of GCR rates in *tel1Δ rad9Δ exo1Δ sgs1Δ* cells expressing either Sgs1, Sgs1$^{APΔ}$, or RBD-Sgs1$^{APΔ}$. Bars represent median values and error bars represent standard deviation from 32 independent colonies. *P* value was calculated using a two-tailed, unpaired *t* test. ****$P \leq 0.0001$. (J) Measurement of GCR rates in *ddc1ΔMAD dna2-aa tel1Δ* cells expressing Sgs1, Dpb11$^{BRCT3/4}$-Sgs1, or RBD-Sgs1. Bars represent median values and error bars represent standard deviation from 32 independent colonies. *P* value was calculated using a two-tailed, unpaired *t* test. *$P \leq 0.05$; ****$P \leq 0.0001$. Source data are available online for this figure.

results suggest that RBD-Sgs1 hinders HR completion presumably by increasing heteroduplex rejection, which delays HR-mediated DNA repair, causing persistent Rad52 foci and stronger genotoxin sensitivity. Next, RBD was fused to the Sgs1$^{APΔ}$ mutant to test the prediction that the high GCR rate observed in Sgs1$^{APΔ}$ was caused by impaired Sgs1 recruitment and that expression of a RBD-Sgs1$^{APΔ}$ should suppress high GCR rates. Indeed, RBD-Sgs1$^{APΔ}$ almost eliminates GCRs in *tel1Δ rad9Δ exo1Δ sgs1Δ* cells (Fig. 3I). To further test the model that GCRs accumulate in cells lacking Mec1 due to the inability of Sgs1 to be properly recruited, we asked whether RBD-Sgs1 can suppress GCRs in Mec1-deficient cells. Since *mec1Δ tel1Δ* cells exhibit limited viability, we opted to use *ddc1Δ dna2-aa tel1Δ* cells expressing the Mec1 activation domain (MAD) of Dna2, which we have previously shown to impair Mec1 signaling and accumulate high GCR rates, while still displaying close to normal growth rates (Lanz et al, 2018) (Appendix Fig. S6). Ectopic expression of wild-type Sgs1 or Dpb11$^{BRCT3/4}$-Sgs1 showed similar GCR rates in *ddc1ΔMAD dna2-aa tel1Δ* cells, consistent with the fact that Dpb11 relies on Ddc1 for proper recruitment (Puddu et al, 2008). In contrast, expression of RBD-Sgs1 fully suppressed GCRs, indicating that engineered Sgs1 recruitment can suppress GCRs in Mec1-deficient cells (Fig. 3J). Overexpression of Sgs1 via a strong *CYC1* promoter could also decrease the GCR rate, but the suppression was not as strong as RBD-Sgs1 (Appendix Fig. S7A,B). We further confirmed that the GCR suppression observed upon RBD-Sgs1 expression is not due to overexpression of the fusion protein since the RBD-Sgs1 fusion was in fact less abundant than Sgs1 (Appendix Fig. S7C,D). Expression of an Ddc1$^{T602A}$-Sgs1 fusion (causing constitutive recruitment to the 911 clamp) could also efficiently suppress GCRs in *dna2-aa ddc1Δ tel1Δ* cells (Appendix Fig. S8), further supporting the model that in cells lacking Mec1, GCRs accumulate due to the inability of Sgs1 to be properly recruited.

Sgs1 is a large multi-domain protein (Appendix Fig. S9A). To define the critical regions required for the GCR suppressive function of Sgs1 when fused to RBD, we generated several mutations and truncations in Sgs1 and monitored GCR rates. Removal of the Top3 interacting motif (TIM, 1–158 aa) did not affect the function of the chimera (Fig. 4A), indicating that the ability of Sgs1 to bind to Top3 was not necessary for GCR suppression when Sgs1 was hyper-recruited. Chimeras with either helicase-defective mutation (*hd*, *SGS1$^{K706A}$*) or deletion of the RQC domain (1081–1195 aa, a region found only in the RecQ helicase family (Huber et al, 2006)) lost the ability to suppress GCRs (Fig. 4A), showing that the helicase activity of Sgs1 is essential for

GCR suppression. Loss of the Helicase and RNaseD C-terminal domain (HRDC, 1271–1351 aa), which is involved in DNA binding (Von Kobbe et al, 2003), caused no change in GCR rate (Fig. 4A), likely due to the fact that RBD can recruit Sgs1 to DNA. Similar results were obtained using *ddc1Δ tel1Δ rad53Δ* cells, among which Sgs1 recruitment is impaired and Mec1 signaling is partially disrupted (Appendix Fig. S10).

Since Mec1 phosphorylation of Sgs1 has recruitment-independent roles (Fig. 2D), we predicted that RBD-Sgs1$^{9mut}$ would partially weaken GCR suppression. We used *ddc1Δ tel1Δ rad53Δ* cells to test this hypothesis because in this strain, endogenous Sgs1 recruitment via 911-Dpb11 is impaired while Mec1 can still be activated via Dna2 and phosphorylate RBD-Sgs1. As expected, loss of Mec1 phosphorylation impaired the suppression of GCRs; however, the change was modest (Fig. 4B). When we introduced serine-to-alanine mutations at other 6 positions, containing 4 putative CDK phosphorylation sites (Appendix Fig. S9B), serine-proline motifs, we also observed a modest increase in GCR rate (Fig. 4B). Strikingly, impairing both Mec1 and CDK phosphorylation motifs in Sgs1 by combining all 15 mutations (RBD-Sgs1$^{15mut}$) led to synergistic effects (Fig. 4B), suggesting that both Mec1 and CDK promote Sgs1's function in GCR suppression.

Next, we asked whether GCRs can be suppressed by other helicases when fused to RBD, or if Sgs1 has unique properties that confer its GCR suppressive function. We fused RBD to other yeast helicases involved in recombination, including Mph1, Pif1, and Rad5 (Piazza et al, 2019), and found that none of them had the ability to prevent GCR accumulation (Fig. 4C). Fusing RBD to BLM, the human ortholog of Sgs1, also could not inhibit GCRs and even showed a higher GCR rate, similar to helicase-dead Sgs1 (Fig. 4A). Addition of the Top3 interacting motif of Sgs1 to RBD-BLM did not alter GCR rates (Fig. 4C). Taken together, our results suggest that engineered Sgs1 recruitment can effectively suppress GCRs, and this function is highly specific to Sgs1.

## Engineered Sgs1 recruitment suppresses HR-driven GCRs and lowers D-loop levels

Since Sgs1 functions at multiple steps in HR, including DNA end resection, heteroduplex rejection and double Holliday junction (dHj) dissolution (Ira et al, 2003; Zhu et al, 2008; Sugawara et al, 2004; Mankouri et al, 2011), we sought to determine which step in HR is impacted by RBD-Sgs1 and likely contributing to the suppression of GCRs. Defects in heteroduplex rejection can give rise to non-allelic HR events (Piazza et al, 2017), while inefficient dHj dissolution can increase

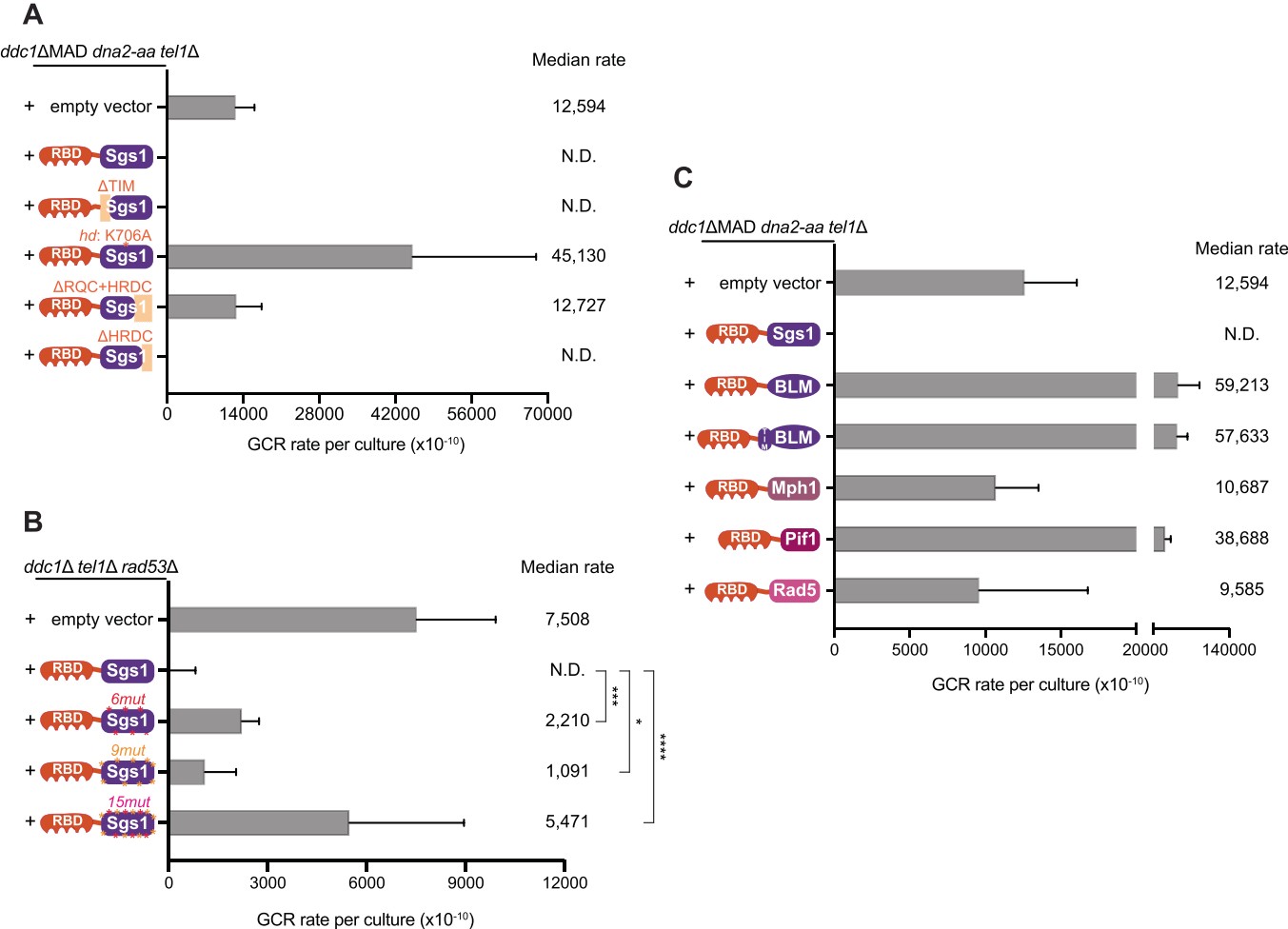

**Figure 4. GCR suppression through the RBD-Sgs1 chimera requires Sgs1 helicase activity and Sgs1 phosphorylation.**

(A) Measurement of GCR rates in *ddc1*ΔMAD *dna2-aa tel1*Δ cells expressing RBD fused to wild-type Sgs1 or truncations of Sgs1 (*hd*: helicase-dead; see legend in Fig. 2A for the description of domains). Bars represent median values and error bars represent standard deviation from 32 independent colonies. (B) Measurement of GCR rates in *ddc1*ΔMAD *dna2-aa tel1*Δ cells expressing RBD fused to wild-type Sgs1 or Sgs1 containing phospho-site mutations (*6mut*: mutation of 6 sites including 4 SP/TP sites; *9mut*: mutation of 9 SQ/TQ sites; *15mut*: combination of *6mut* and *9mut* mutations). Bars represent median values and error bars represent standard deviation from 32 independent colonies. P value was calculated using a two-tailed, unpaired *t* test. *P ≤ 0.05; ***P ≤ 0.001; ****P ≤ 0.0001. (C) Measurement of GCR rates in *ddc1*ΔMAD *dna2-aa tel1*Δ cells expressing RBD fused to yeast DNA helicases Sgs1, Mph1, Pif1 or Rad5, or fused to BLM, the human ortholog of Sgs1. Bars represent median values and error bars represent standard deviation from 16 independent colonies. Source data are available online for this figure.

the occurrence of crossovers (Ira et al, 2003). Defects in both of these processes can induce chromosomal rearrangements. Multiple lines of evidence support the hypothesis that RBD fusion enhances the capacity of Sgs1 to reject heteroduplexes, thereby preventing GCRs driven by non-allelic HR. First, we showed that removal of the TIM of Sgs1 does not affect the ability of RBD-Sgs1 to suppress GCR (Fig. 4A), excluding the requirement for the strand passage function of Top3 that is needed for joint molecule dissolution (Mankouri et al, 2011; Cejka et al, 2012). Second, RBD-Sgs1 could still suppress GCRs in *rad51*Δ cells (Fig. 5A; Appendix Fig. S11A), where Holliday junctions do not form although Rad52-dependent single-strand annealing can still be used (Ivanov et al, 1996), which would necessitate heteroduplex rejection for HR quality control. In this context, if RBD-Sgs1 suppresses GCRs by promoting heteroduplex rejection, RBD-Sgs1 should fail to suppress GCRs in *rad52*Δ cells. Indeed, in the absence of Rad52, the effect of RBD-Sgs1 expression was comparable to that of Sgs1 expression (Fig. 5B; Appendix

Fig. S11B). To directly monitor the effect of RBD-Sgs1 in heteroduplex rejection, we performed the displacement loop (D-loop) capture (DLC) assay (Fig. 5C and (Piazza et al, 2019; Reitz et al, 2022)), which is able to quantify the kinetics of nascent D-loop intermediate formation in vivo and distinguishes itself from other assays limited to measuring the final physical or genetic repair end product. This analysis expands the genetic endpoint analysis in the GCR assay and directly pinpoints the affected HR intermediate. In the DLC assay, D-loop formation between an HO-induced DSB site on chromosome V with a 2 KB homology donor located on chromosome II is physically monitored. Two EcoR1 sites are strategically utilized for proximity ligation, as depicted in (Fig. 5C and (Piazza et al, 2019; Reitz et al, 2022)). Upon GAL-induced HO nuclease expression and subsequent DNA strand invasion at the donor site, psoralen is used for in vivo inter-strand DNA crosslinking, forming covalent links within the heteroduplex DNA (hDNA) within the D-loop, thereby preserving it during subsequent steps. To restore the restriction

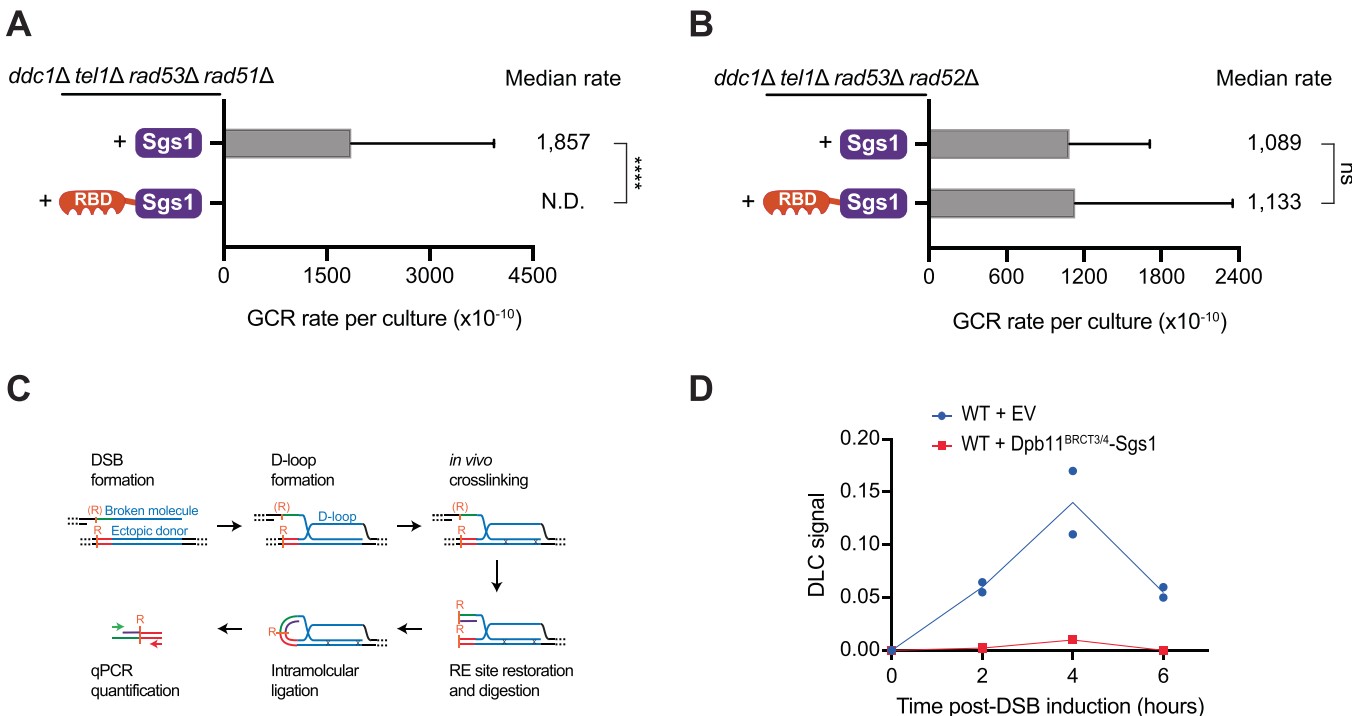

**Figure 5.  Engineered Sgs1 recruitment via RBD-Sgs1 chimera suppresses HR-driven GCRs and lowers D-loop levels.**

(A) Measurement of GCR rates in *ddc1Δ tel1Δ rad53Δ rad51Δ* cells expressing either Sgs1 or RBD-Sgs1. Bars represent median values and error bars represent standard deviation from 32 independent colonies. *P* value was calculated using a two-tailed, unpaired *t* test. ****$P \leq 0.0001$. (B) Measurement of GCR rates in *ddc1Δ tel1Δ rad53Δ rad51Δ* cells expressing either Sgs1 or RBD-Sgs1. Bars represent median values and error bars represent standard deviation from 32 independent colonies. *P* value was calculated using a two-tailed, unpaired *t* test. (C) Schematic representation of the D-loop capture (DLC) assay (Piazza et al, 2019). (D) DLC signal kinetics in cells carrying an empty vector or expressing the DPB11^BRCT3/4-Sgs1 chimera. Source data are available online for this figure.

site that was ablated by DNA end resection, a long complementary oligonucleotide is introduced. Following restoration of the restriction site and its digestion, the crosslinked hDNA is selectively coupled during the proximity ligation reaction. The resulting distinct chimeric ligation products are quantified via quantitative PCR (qPCR) using a pair of specific primers, yielding the DLC signal. A decreased DLC signal shows lower D-loop levels and indicates stronger heteroduplex rejection. Wild-type cells with empty vector show D-loop levels and D-loop kinetic consistent with previous observations (Piazza et al, 2019; Reitz et al, 2022), while cells expressing Dpb11^BRCT3/4-Sgs1 and RBD-Sgs1 show strongly diminished D-loop levels. These results are consistent with the hypothesis that the engineered Sgs1 recruitment suppresses GCR by promoting heteroduplex rejection through D-loop disruption (Fig. 5D; Appendix Fig. S12A–C). Taken together, our results show that engineered Sgs1 recruitment suppresses HR-driven GCRs through enhanced heteroduplex rejection.

## The helicase activity of Sgs1 and its phosphorylation by Mec1 suppress homeologous recombination

While our findings show that the ability of the RBD-Sgs1 chimera to suppress GCRs is associated with a strong heteroduplex rejection activity, the engineered system for Sgs1 recruitment and the DLC assay have limitations for probing the action and regulation of Sgs1 in heteroduplex rejection. In particular, the DLC assay currently available only monitors D-loops formed by homologous sequences,

and therefore cannot be used to assess how the Mec1-Sgs1 axis acts on heteroduplexes formed between homeologous sequences. This is particularly relevant since Sgs1 acts preferentially to suppress homeologous recombination (Myung et al, 2001b). Moreover, the helicase activity of Sgs1 was found to be dispensable for the disruption of D-loops between homologous sequences (Piazza et al, 2019), consistent with the model that Sgs1 disrupts D-loops between homeologous sequences, which offers an attractive explanation for how Sgs1, and its phospho-regulation by Mec1, suppress non-allelic recombination and GCRs.

To test the model that Sgs1 and its regulation by Mec1 phosphorylation are important for counteracting homeologous recombination, we performed a single-strand annealing (SSA) assay that is based on homologous (0% sequence divergence, AA) or homeologous (3% sequence divergence, FA) sequences (Fig. 6A and (Sugawara et al, 2004)). A higher ratio of AA repair efficiency to FA repair efficiency suggests stronger rejection of homeologous recombination. In *rad9Δ* cells, deletion of Sgs1 decreases the AA/FA ratio to 1 (Fig. 6B–D), indicating that Sgs1 is required to distinguish between homologous from homeologous sequences and to prevent homeologous recombination, as previously shown (Sugawara et al, 2004; Goldfarb and Alani, 2005). In *rad9Δ* cells expressing Sgs1^hd, the ability to reject homeologous recombination was strongly reduced (Fig. 6B–D), consistent with the prediction that the helicase activity of Sgs1 is important for the rejection of heteroduplexes between homeologous sequences. Consistent with

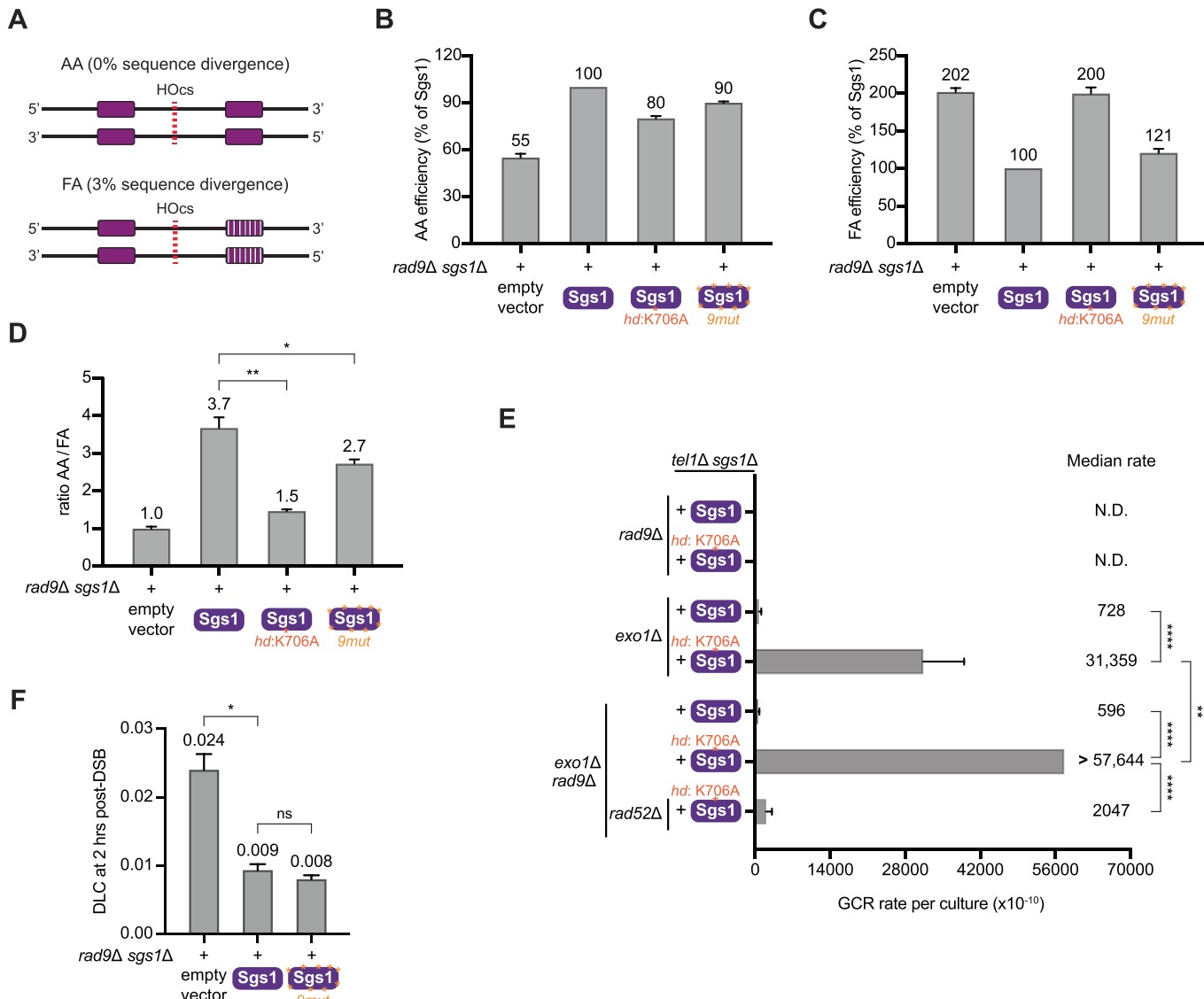

**Figure 6. Sgs1 helicase activity and its phosphorylation by Mec1 are both important for the rejection of homeologous recombination.**

(A) Diagram of the single-strand annealing (SSA) assay used in this study (Sanford et al, 2021). Red line represents a galactose-inducible HO endonuclease cut site. AA strain contains perfect homology on either side of a GAL-inducible break. FA strain contains 3% sequence divergence on either side of the break (homeologous recombination). (B) Measurement of SSA efficiency in the AA strains expressing an empty vector or Sgs1 mutants. Bars represent mean values and error bars represent standard error of the mean from four replicate experiments. (C) Measurement of SSA efficiency in the FA strains expressing an empty vector or Sgs1 mutants. Bars represent mean values and error bars represent standard error of the mean from four replicate experiments. (D) AA/FA ratio computed from data in (B, C). P value was calculated using a two-tailed, unpaired t test. *P ≤ 0.05; **P ≤ 0.01. (E) Measurement of GCR rates in cells with the indicated genotypes expressing either Sgs1 or Sgs1[hd]. Bars represent median values and error bars represent standard deviation from 32 independent colonies. "N.D." indicates "not detected". P value was calculated using a two-tailed, unpaired t test. **P ≤ 0.01; ****P ≤ 0.0001. (F) DLC signal in rad9Δ sgs1Δ cells expressing either empty vector, Sgs1, or Sgs1[9mut]. Bars represent mean values and error bars represent standard error of the mean from three replicate experiments. P value was calculated using a two-tailed, unpaired t test. *P ≤ 0.05. Source data are available online for this figure.

this finding and with the model that GCRs are strongly induced by homeologous recombination in conditions of deregulated resection, mutation of the helicase domain of Sgs1 resulted in drastic increases in GCRs rates (Fig. 6E). Interestingly, expression of Sgs1[9mut] significantly decreases the AA/FA ratio (Fig. 6B–D), indicating that the phosphorylation of Sgs1 by Mec1 contributes to the ability of Sgs1 to reject homeologous recombination. Expression of Sgs1[9mut] did not have a significant impact in the

formation of D-loops formed between homologous sequences (Fig. 6F), further suggesting that the phosphorylation of Sgs1 by Mec1 is specifically important for the rejection of homeologous heteroduplexes. The ~2.5-fold effect of Sgs1 on D-loop levels is consistent with previous observations (Piazza et al, 2019). We note that the effect of Mec1 phosphorylation on the function of Sgs1 is, at least partially, independent of Msh2 with regards to GCR suppression (Appendix Fig. S13A), consistent with previous

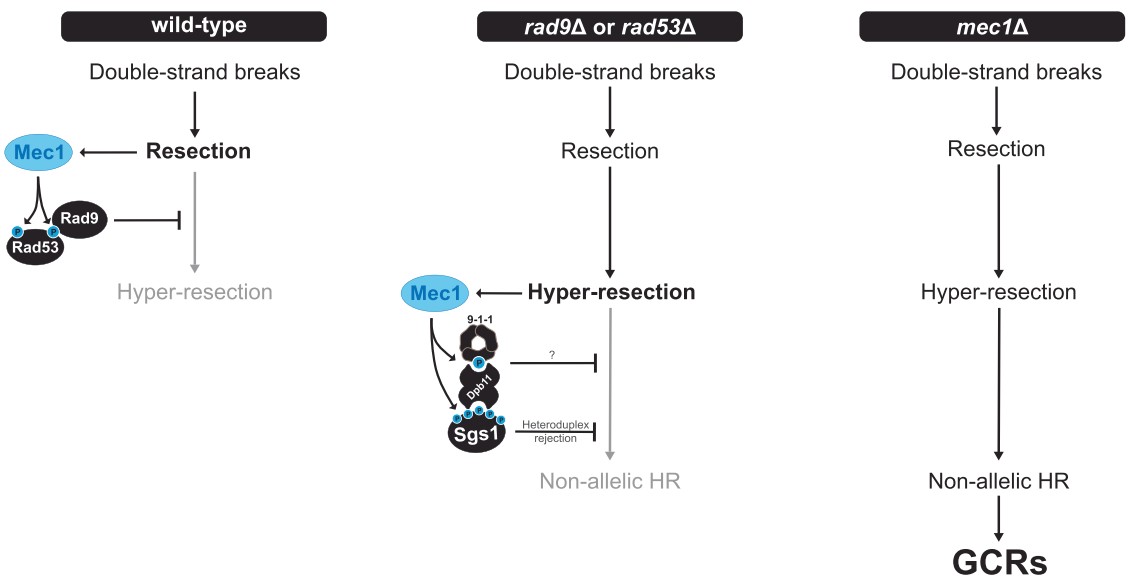

**Figure 7. Model for GCR suppression via multi-step control of HR by Mec1.**

Upon DSB and initial end resection, Mec1 is recruited to RPA-ssDNA to promote the Rad9-Rad53 signaling axis that restrains long-range resection. This anti-resection function of Mec1 protects DNA ends from extensive nucleolytic processing, thereby reducing the chance of non-allelic HR and preventing GCRs. In cells lacking *RAD9* or *RAD53*, DNA ends undergo hyper-resection, which activates a mode of Mec1 signaling leading to Sgs1 phosphorylation, and its recruitment to lesion sites via the 911-Dpb11 complex. The phosphorylation of Sgs1 by Mec1 boosts the ability of Sgs1 to inhibit non-allelic HR through the rejection of heteroduplexes between homeologous sequences, thereby suppressing GCRs. Mec1 phosphorylation of the 9-1-1 clamp at residue T602 of the Ddc1 subunit is also important for GCR suppression through the recruitment of Sgs1 and through yet unknown mechanism(s) that are independent of Sgs1 phosphorylation. Not depicted in the schematic, Mec1 also phosphorylates additional targets to promote GCR suppression since the GCRs rates of cells lacking *SGS1* and *RAD53* are lower than the GCR rates of cells lacking *MEC1*. *mec1Δ* cells fail to restrain resection and also lack the Mec1-Sgs1 salvage pathway (impaired HR quality control), leading to a dramatic increase of non-allelic HR-driven GCRs. Also not depicted in the schematic, in the absence of Mec1, Tel1 plays a crucial role in suppressing a massive increase in GCR rates, possibly by promoting the repair of the extensive number of DSBs that accumulate.

observations that Sgs1 and Msh2 can contribute to suppressing GCRs driven by homeologous recombination through parallel pathways (Myung et al, 2001b). In further support of this notion, we found that Sgs1 contributes to the rejection of homeologous recombination in *msh2Δ* cells in a manner that depends on the 9S/T–Q sites mutated in Sgs1[9mut] (Appendix Fig. S13B–D). Moreover, while the lack of Sgs1 completely impairs any preferential rejection of homeologous recombination, some level of biased rejection of homeologous recombination can still be observed in cells lacking Msh2 (Appendix Fig. S13B–D), pointing to Msh2-independent roles for Sgs1 in the rejection of homeologous recombination. Taken together, our results are consistent with the model that Mec1 suppresses GCRs, in part, by phosphorylating Sgs1 to promote the rejection of homeologous recombination.

## Discussion

Over 20 years ago, foundational work by the group of Richard Kolodner revealed elevated rates of GCRs in *mec1Δ tel1Δ* cells (Myung et al, 2001c). Their work also showed that the ability of Mec1 and Tel1 to suppress GCRs is largely independent of their canonical role in activating the DNA damage checkpoint (Myung et al, 2001c; Lanz et al, 2018). The detailed mechanism by which Mec1 and Tel1 suppress GCRs has remained elusive, representing a major gap in our understanding of kinase-mediated genome maintenance mechanisms. Here we focused on the GCR

suppressive function of Mec1 and found that *mec1Δ* cells accumulate GCRs that are driven by deregulated HR. Moreover, we revealed that higher GCR rates are caused by compounding effects from the combined loss of DNA damage checkpoint and the control of Sgs1 (Fig. 7). Our findings show that, upon loss of DNA damage checkpoint signaling and resection control, a Mec1-Sgs1 salvage pathway limits GCR accumulation. We propose that this salvage pathway increases the rejection of homeologous recombination, functioning as a boosted HR quality control mechanism that limits non-allelic recombination.

Quantitative phosphoproteomic analysis of *rad53Δ* or *rad9Δ* cells showed that these mutants display increased Mec1 signaling directed towards a selective group of proteins involved in ssDNA transactions. In particular, the hyperphosphorylation of the Sgs1 helicase in these strains promotes its recruitment to DNA lesion sites via the association with the 911-Dpb11 complex (Sanford et al, 2021). The discovery of novel modes of Mec1/ATR signaling upon loss of checkpoint reveals the multi-faceted action and complex regulation of this kinase. Since *rad9Δ* cells do not suffer the drastic replication fork collapse phenotype observed in *rad53Δ* cells, we favor the model that the hyperactivation of Mec1 observed in both *rad53Δ* or *rad9Δ* cells is caused by deregulated resection. Notably, the lack of Rad53 has been shown to impair Rad9's role in counteracting resection (Gobbini et al, 2015; Clerici et al, 2014), consistent with both *rad53Δ* or *rad9Δ* cells sharing a similar defect in resection control. Exactly how deregulated resection promotes Mec1 signaling is still unclear. One possibility is that faster rates of

resection, or imbalanced engagement of resection nucleases Exo1 and Dna2, causes abnormal exposure of ssDNA that is sensed by Mec1. Since increased exposure of ssDNA is expected to increase non-allelic recombination, an interesting implication of our model is that the signal for Mec1 activation is the actual driver of GCR events, implying that Mec1 signaling serves as a rheostat to increase heteroduplex rejection and HR quality control. We propose that tightly controlling heteroduplex rejection in a context-dependent manner, and not overstimulating it when not needed, is crucial to make sure HR can be properly utilized for DNA repair transactions, such as template switching, when needed. Moreover, our findings, and previous reports (Doerfler et al, 2011), highlight an important role for Exo1 in preventing GCRs and that *exo1Δ* cells have an increased demand for Sgs1 regulation. Whereas Exo1 and Dna2-Sgs1 are involved in extensive resection (Zhu et al, 2008; Mimitou and Symington, 2009), Rad9 was reported to prevent hyper-resection by Sgs1 (Leland et al, 2018), with faster resection in *rad9Δ* cells being mainly dependent on Sgs1 (Bonetti et al, 2015). Thus, Exo1 may play an important role in competing with Dna2-Sgs1, which may ensure proper resection.

Our results suggest that Mec1 controls the function of Sgs1 through at least two mechanisms (Fig. 7, middle panel). In addition to regulating Sgs1's recruitment via the 911-Dpb11 complex, as we previously showed (Sanford et al, 2021), Mec1 also promotes Sgs1's function in inhibiting homeologous recombination via direct phosphorylation. These two outcomes of Mec1 signaling may be interdependent, with Mec1 phosphorylating Ddc1 at threonine 602 to provide a docking site for Dpb11, which contributes to bridging Sgs1 to sites nearby ssDNA and closer to Mec1, therefore enhancing the phosphorylation of Sgs1 by Mec1. Notably, Sgs1 may also be directly recruited to ssDNA via its direct interaction with RPA (Hegnauer et al, 2012). Both modes of recruitment may contribute to the ability of Sgs1 to reject homeologous recombination, with the direct phosphorylation of Sgs1 by Mec1 serving as an additional mode of regulation.

While we cannot rule out a potential role for the Mec1 signaling in affecting the role of Sgs1 in DNA end resection, several lines of evidence disfavor this notion, and support that Mec1 controls the heteroduplex rejection function of Sgs1. First, we observed no changes in checkpoint signaling upon expression of RBD-Sgs1, suggesting that there are no major changes in ssDNA exposure caused by RBD-Sgs1. Second, the RBD-Sgs1 chimera drastically reduces GCR as opposed to increasing GCRs. If RBD-Sgs1 would result in increased resection, the expectation would be more ssDNA exposure and increased rates of GCRs. Third, in the absence of Exo1, the Sgs1^9mut resulted in more GCRs, which is not consistent with the expected outcome of reduced resection. Fourth, Sgs1^9mut is at least partially epistatic with Msh2 deletion, a protein involved in the rejection of homeologous heteroduplexes and with no documented roles in DNA end resection. Fifth, we inferred from genetic endpoint assays that Sgs1 acts on the D-loop pairing intermediate and directly showed that Sgs1 chimera lowers D-loop levels using a recently developed physical assay for D-loops.

How Sgs1 rejects heteroduplexes between homeologous sequences is a key outstanding question, with implications to defining the precise molecular mechanism of how Mec1 signaling regulates HR and GCR suppression. It is surprising that the ability of Sgs1 to suppress homeologous recombination is, at least partially, independent of the mismatch repair protein Msh2. In

further support of this notion, we found that Sgs1 contributes to the rejection of homeologous recombination in *msh2Δ* cells in a manner that depends on the Mec1 phosphorylation. Consistent with our findings, a previous study showed that Sgs1 and Msh2 have synergistic effects in the suppression of GCRs and homeologous recombination, and operate through at least partially independent mechanisms (Myung et al, 2001b). Furthermore, *sgs1Δ* cells have higher GCR rates and more homeologous translocation events compared to *msh2Δ* cells (Myung et al, 2001b). Future work should focus on understanding how Sgs1 distinguishes homeologous sequences independently of Msh2. In addition, it will be important to investigate how Mec1 phosphorylation modulates the helicase activity of Sgs1 and its ability to reject homeologous sequences. We have shown that RBD-Sgs1 requires phosphorylation at both CDK and Mec1 sites to efficiently prevent GCR accumulation. Notably, CDK phosphorylation sites on Sgs1 have been shown to stimulate DNA unwinding (Grigaitis et al, 2020). Thus, it's tempting to propose that Mec1 regulates the helicase activity of Sgs1 via phosphorylation to enhance its heteroduplex rejection function and suppress GCRs.

Our findings also point to interesting features of Sgs1 action and regulation that seem unique to this helicase in GCR suppression. For example, we show that fusing other yeast helicases to the RBD domain does not result in appreciable GCR suppression as seen with the RBD-Sgs1 chimera, potentially due to the fact that these other helicases do not share a key protein interaction(s) and/or display the same ability of Sgs1 to efficiently reject heteroduplexes between homeologous sequences. In the case of BLM, we excluded the possibility that a lack of interaction with yeast Top3 was the cause of its inability to suppress GCRs since expression of a RBD-TIM-BLM chimera, which should promote interaction of this BLM fusion protein to Top3, generated even more GCRs, with values similar to that of the RBD-Sgs1^hd. Therefore, expression of BLM may lack specific regulatory mechanisms (such as phosphorylation) and/or yet unknown interaction(s) that promote the rejection of homeologous recombination. Thus, similar to RBD-Sgs1^hd, RBD-TIM-BLM may be able to disrupt homologous D-loops but fails to efficiently disrupt D-loops between homeologous sequences, leading to more GCRs.

Mec1 is expected to suppress GCRs through additional mechanisms that do not require Sgs1, as evidenced by a comparison of GCR rates in different mutant strains. For example, the GCR rates of *tel1Δ rad9Δ exo1Δ* Sgs1^9mut Ddc1^T602A (~11,000) are significantly lower compared to that of *mec1Δ tel1Δ* (~45,000). Consistent with Sgs1-independent roles for Mec1 in GCR suppression, our phosphoproteomic analysis revealed that loss of *RAD9* or *RAD53* induces phosphorylation of other proteins with roles in ssDNA-associated transactions, such as Rfa2 and the ubiquitin ligase and DNA translocase Uls1. Further dissecting the roles of these, and potentially other, Mec1 phosphorylation events induced in *rad9Δ* cells should shed light into additional GCR suppressing mechanisms controlled by Mec1. Moreover, it will be important to define the role of Tel1 in limiting GCR accumulation upon loss of Mec1. One possibility is that DSBs accumulate in *mec1Δ* cells due to increased fork collapse, and that Tel1 is required to properly repair these breaks and prevent them from engaging in deleterious DNA transactions that cause GCRs.

Although this work addresses how Mec1 prevents non-allelic HR-driven GCRs, it is worth mentioning that GCRs can also arise

by HR-independent pathways, such as de novo telomere addition and non-homologous end joining (NHEJ) (Myung et al, 2001a; Putnam and Kolodner, 2017). This explains how GCRs accumulate in *rad52Δ* cells that are unable to use HR, and why RBD-Sgs1 fails to suppress these GCR events.

Whereas yeast offers a robust and much-simplified system to dissect mechanisms of GCR suppression, we envision that our findings may contribute to better understanding GCR suppression mechanisms in mammals. For example, exploring how mammalian cells respond to deregulated resection may uncover similar salvage pathways involved in heteroduplex rejection control as the Mec1-Sgs1 pathway identified here. Interestingly, BLM has been shown to interact with TOPBP1, the ortholog of Dpb11, although the interaction is not dependent on ATR (Blackford et al, 2015; Sun et al, 2017; Balbo Pogliano et al, 2022). Nevertheless, BLM is phosphorylated by ATR (Davies et al, 2004), which could have an effect on BLM's function in heteroduplex rejection. It is also possible that ATR may respond to deregulated resection in a more complex manner than Mec1 does in yeast, involving a larger set of substrates and GCR suppression mechanisms. Moreover, it is also possible that ATR-independent responses are triggered upon deregulated resection and actively control heteroduplex rejection to limit genetic instability. In summary, exploring the response to deregulated resection in mammals may open new directions to understand mechanisms of genome maintenance.

# Methods

## Yeast strains

A complete list of yeast strains used in this study can be found in Table EV1. The strain background for all yeast used in this study is S288C, unless indicated. Gene deletions were performed using standard PCR-based strategy to amplify resistance cassettes with flanking sequences homologous to the target gene. All endogenous deletions were verified by PCR. Plasmids in this study are listed in Table EV2 and are available upon request. Yeast strains were grown at 30 °C in a shaker at 220 rpm. For strains with endogenous deletion, YEPD media were used. For strains carrying plasmids, the corresponding synthetic dropout media were used. For SILAC experiments, yeast strains were grown in -Arg -Lys media supplemented with either isotopically normal arginine and lysine ("light" media) or the $^{13}C^{15}N$ isotopologue ("heavy" media). Excess proline was added to SILAC media at a concentration of 80 mg/L to prevent conversion of arginine to proline.

## Western blots

In all, 50 ml of yeast were grown in appropriate media to mid-log phase and treated as described in the figure legend. Cells were pelleted at 1000 rcf and washed with TE buffer (pH 8.0) containing 1 mM PMSF. Pellets were lysed by bead beating with 0.5-mm glass beads for three cycles of 10 min with 1 min rest time between cycles at 4 °C in lysis buffer (150 mM NaCl, 50 mM Tris pH 8.0, 5 mM EDTA, 0.2% Tergitol type NP-40) supplemented with complete EDTA-free protease inhibitor cocktail (Roche), 1 mM PMSF, 5 mM sodium fluoride, and 10 mM b-glycerophosphate. Concentration normalization was performed via the Bradford assay. Lysates were

boiled in Laemmli buffer and electrophoresed on a 10% SDS–PAGE gel. Proteins were then transferred wet onto a PVDF membrane and incubated with antibody. Signal detection was performed using HRP-coupled secondary antibodies, imaged with BioRad ChemiDoc.

## Phosphoproteomics

For phosphoproteomic experiments, 150 ml of yeast were grown in "heavy" or "light" SILAC media to mid-log phase and treated with 0.04% MMS for 2 h. Cells were pelleted and lysed as described for western blots above. Protein digestion, phosphoenrichment, and following MS data analysis were performed as described in Sanford et al, 2021.

## Immunoprecipitation–mass spectrometry (IP-MS)

For IP-MS experiment, 150 ml of yeast were grown in "heavy" or "light" SILAC media to mid-log phase. Cells were pelleted and lysed as described for western blots above. Around 5 mg of lysate per sample was incubated with antibody-conjugated agarose resin (Anti-c-Myc, Sigma) for 3 h at 4 °C. Resin was washed four times in the lysis buffer. Proteins were eluted by heating at 65 °C with elution buffer (1% SDS, 100 mM Tris pH 8.0) for 15 min. MS samples preparation were performed as described in Sanford et al, 2021.

## GCR assay

All GCR assays were performed with yeast freshly streaked from frozen glycerol stocks or new transformations. Plates were incubated at 30 °C for 3–4 days to get visible colonies. Individual colonies with similar sizes were picked and transferred to 2 ml of culture (YPD for strains with integrated genetic modification, -Leu media for strains with pRS415 plasmids). After 48 h, ~10 million cells were spun down, washed with 400 µl of autoclaved ddH$_2$O, resuspended in 150–200 µl of autoclaved ddH$_2$O and spotted onto plates containing 5-FOA and canavanine (Putnam and Kolodner, 2010). Fewer cells were used when strains have extremely high GCR rates, e.g., *exo1Δ sgs1Δ*. In parallel with each GCR experiment, multiple cultures (usually 4 in this study) were randomly chosen and serially diluted (for YPD, $2 × 10^6$; for -Leu, $5 × 10^4$) and plated onto YPD plates to determine the average population viability. After 4 days, the number of 5-FOA- and canavanine-resistant colonies in a spot was counted. The number of GCR events in culture was calculated using the equation $m[1.24 + \ln(m)] − r = 0$, where r is the number of 5-FOA- and canavanine-resistant colonies in a spot, and m is the estimated number of GCR events (Putnam and Kolodner, 2010). GCR rate was then calculated by dividing the number of GCR events per culture by the average population viability. For each GCR experiment, at least 16 independent colonies were picked, and 2 independent strains with the same genotype were used.

## SSA assay

All SSA assays were performed with yeast freshly streaked from frozen glycerol stocks or new transformations. Plates were incubated at 30 °C for 3–4 days to get visible colonies. Individual

colonies with similar sizes were picked and transferred to 2 ml of culture (YPD for strains with integrated genetic modification, -Leu media for strains with pRS415 plasmids). After 48 h, cells were diluted to 1 $OD_{600}$/ml and ~200 cells were plated onto YP glucose plates (-Leu plates for strains with pRS415 plasmids), ~400 cells were plated onto YP galactose plates. Plates were incubated at 30 °C for 4 days prior to counting colonies. For each isolate, SSA efficiency was calculated by dividing the number of colonies on galactose plates by two times the number of colonies on glucose plates. SSA efficiency of AA strain was divided by the SSA efficiency of FA strain to get a measure of heteroduplex rejection. Four biological replicates were performed for each strain and each biological replicate contained three independent isolates.

### D-loop capture assay

For D-loop capture experiments, all strains were in the W303 RAD5 background. They contain a copy of the GAL1/10-driven HO endonuclease gene at the TRP1 locus on chr. IV. A point mutation inactivates the HO cut site at the mating-type locus (MAT) on chr. III (MATa-inc). The DSB-inducible construct contains the 117 bp HO cut site, a 2086 bp-long homology A sequence (+4 to +2090 of the LYS2 gene), and a 327 bp fragment of the PhiX174 genome flanked by multiple restriction sites (Piazza et al, 2019). D-loop capture assay was performed as previously reported D-loop capture assay was performed as previously reported along with the controls monitoring DSB formation, ligation efficiency, and a normalization locus (Appendix Fig. S12B,C) (Piazza et al, 2019; Reitz et al, 2022), with the following modifications: zymolyase lysed cells were proceeded immediately to the restriction digestion, ligation and DNA purification step after hybridization with oligonucleotides as described previously (Piazza et al, 2018).

### Microscopy analysis

For Rad52 foci analysis, cells were grown at 30 °C in synthetic complete media (for rad9Δ and rad53Δ microscopy) or -Leu media (for Sgs1 rejector microscopy) until $OD_{600}$ reaches 0.2, and 0.01% MMS was added to the culture for 2 h if mentioned. Next, 200 μl of culture was transferred to 4-chamber glass bottom dishes (Cellvis), which were pre-treated with 0.5 mg/ml concanavalin A (Sigma). After 5 min of fixation, liquid was aspirated, and cells were washed with 200 μl of autoclaved ddH2O. 1 ml of requisite media was added to keep cells alive during imaging. Over 150 cells were scored for each replicate. Images were acquired at room temperature using a spinning-disc confocal microscope (CSU-X; Yokogawa Electric Corporation and Intelligent Imaging Innovations) on an inverted microscope (DMI600B; Leica Biosystems) with a 100×, 1.46 NA objective lens and an electron-multiplying charge-coupled device camera (QuantEM; Photometrics). 488 nm laser lines were used for the detection of mRuby-tagged Rad52 in yeast cells. SlideBook software (Intelligent Imaging Innovations) was used to obtain Z stack images. Maximum intensity projections were created in the Slidebook software for foci number analysis.

### Dilution assays

For dilution assays, 3 ml of yeast culture was grown to saturation at 30 °C. Then, 1 $OD_{600}$ equivalent of the saturated culture was serially diluted (tenfold serial dilutions were used unless noted) in a 96-well plate with autoclaved ddH2O and spotted onto agar plates using a bolt pinner. Plates were incubated at 30 °C for 2 days before imaging.

## Data availability

The mass spectrometry data from this publication have been deposited to the PRIDE database (https://www.ebi.ac.uk/pride/archive/) and assigned the identifiers PXD051892.

The source data of this paper are collected in the following database record: biostudies:S-SCDT-10_1038-S44318-024-00139-9.

## Peer review information

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

## Acknowledgements

The authors thank Beatriz S Almeida for technical support; we thank members of the Smolka Lab for valuable discussions related to this work. This work is supported by grants from the National Institute of Health, R35GM141159 to MBS, and R01GM58015 and R01GM137751 to WDH. SHH was partially supported by a fellowship from the Academia Sinica, Taiwan.

## Author contributions

**Bokun Xie**: Conceptualization; Data curation; Formal analysis; Investigation; Methodology; Writing—original draft; Writing—review and editing. **Ethan James Sanford**: Data curation; Writing—review and editing. **Shih-Hsun Hung**: Data curation; Writing—review and editing. **Mateusz Wagner**: Data curation; Writing—review and editing. **Wolf-Dietrich Heyer**: Data curation; Supervision; Funding acquisition; Methodology; Writing—review and editing. **Marcus B Smolka**: Conceptualization; Data curation; Supervision; Funding acquisition; Writing—original draft; Project administration; Writing—review and editing.

Source data underlying figure panels in this paper may have individual authorship assigned. Where available, figure panel/source data authorship is listed in the following database record: biostudies:S-SCDT-10_1038-S44318-024-00139-9.

## Disclosure and competing interests statement

The authors declare no competing interests.

