## [Peer Review File · The EMBO Journal]

Multi-Step Control of Homologous Recombination Via Mec1/ATR Suppresses Chromosomal Rearrangements

Bokun Xie, Ethan Sanford, Shih-Hsun Hung, Mateusz Wagner, Wolf-Dietrich Heyer, and Marcus Smolka

Corresponding author(s): Marcus Smolka (mbs266@cornell.edu)

Review Timeline:

Submission Date:	4th Dec 23
Editorial Decision:	15th Jan 24
Revision Received:	29th Apr 24
Editorial Decision:	17th May 24
Revision Received:	19th May 24
Accepted:	22nd May 24

Editor: Hartmut Vodermaier

Transaction Report:

Dr. Marcus B Smolka
Cornell University
Weill Institute for Cell and Molecular Biology
339 Weill Hall
Ithaca, NY 14853-7202

15th Jan 2024

Re: EMBOJ-2023-116302
Multi-Step Control of Homologous Recombination Via Mec1/ATR Suppresses Chromosomal Rearrangements

Dear Marcus,

Thank you for submitting your study on HR control by Mec1 and Sgs1 to The EMBO Journal. I have now received reviews from three expert referees, copied below for your information. As you will see, all referees appreciate the potential interest of your findings, but also raise a number of issues that would need to be satisfactorily addressed prior to publication. In particular, referee 3 brings up several substantive concerns, one of which is also echoed by referee 1.

Should you be able to adequately respond to these criticisms, we would be happy to consider a revised manuscript further for publication. Since it is our policy to consider only a single round of major revision, it is important to fully answer to all comments at the time of resubmission - I would therefore invite you to get back to me with a tentative response letter/revision plan already during the early stages of the revision work. On the basis of this response, we could then have a call in which we may further discuss the revision requirements and how to best address the key issues. I should add that we could also offer extension of the default three-months revision period if needed, with our 'scooping protection' (meaning that competing work appearing elsewhere in the meantime will not affect our considerations of your study) remaining of course valid also throughout this extension.

Detailed information on preparing, formatting and uploading a revised manuscript can be found below and in our Guide to Authors. Thank you again for the opportunity to consider this work for The EMBO Journal, and I look forward to hearing from you in due time.

With kind regards,

Hartmut

9) Digital image enhancement is acceptable practice, as long as it accurately represents the original data and conforms to community standards. If a figure has been subjected to significant electronic manipulation, this must be clearly noted in the figure legend and/or the 'Materials and Methods' section. The editors reserve the right to request original versions of figures and the original images that were used to assemble the figure. Finally, we generally encourage uploading of numerical as well as gel/blot image source data; for details see: embopress.org/page/journal/14602075/authorguide#sourcedata

At EMBO Press, we ask authors to provide source data for the main manuscript figures. Our source data coordinator will contact you to discuss which figure panels we would need source data for and will also provide you with helpful tips on how to upload and organize the files.

In the interest of ensuring the conceptual advance provided by the work, we recommend submitting a revision within 3 months (14th Apr 2024). Please discuss the revision progress ahead of this time with the editor if you require more time to complete the revisions. Use the link below to submit your revision:

Link Not Available

Referee #1:

The authors use genetics of the budding yeast as a model system to study pathways leading to the suppression of gross chromosomal rearrangements (GCR). The canonical pathway involves the Mec1 kinase and the signaling cascade downstream (Rad53/Rad9). In the absence of this axis, the authors uncover a salvage pathway involving the Sgs1 helicase. The study then narrows down to define the function of Sgs1 in GCR suppression, which is particularly apparent in cells lacking Rad53 and/or Rad9 (checkpoint proteins) or Exo1 (end resection nuclease). Sgs1 function in this context is stimulated by Mec1 phosphorylation (of SQ/TQ sites), but also by CDK phosphorylation. Sgs1 fused to an RPA-binding domain facilitates its function in GCR suppression, and can partially bypass the role of Mec1 phosphorylation. The authors propose that Sgs1 function in GCR suppression likely involves its function in heteroduplex rejection.

Overall, this is a high-quality and interesting study. The data are described well, especially considering the high complexity of the subject. Although one could argue that the Sgs1 pathway may not be very active in wild type cells, it is important to understand also backup pathways, as they will guide future research in human cells. Therefore, I believe the manuscript addresses an important topic.

Specific comments:

The salvage pathway preventing GCRs, studied here, involves Sgs1 function that is proposed to be dependent on its role in heteroduplex rejection (known from previous literature). However, whether it is indeed heteroduplex rejection remains somewhat undefined, which represents a weak point of this study.

The authors note that the canonical (in wild type cells) function of Mec1 in the suppression of GCRs involves Rad53 and its function in resection control, and cite relevant literature. The model that the Sgs1 backup pathway functions through heteroduplex rejection is reasonable, but its support is rather indirect (e.g. the function does not involve Top3). Sgs1 also functions in resection together with Dna2. Does the resection role of Sgs1 play any role in this context? The authors could try eliminating Dna2 (or its nuclease activity, with the necessary pif1-m2 suppressor mutation), which would exclude a resection function. On the other hand, testing a genetic interaction of the sgs1 mutation(s) in combination with e.g. msh2-d could provide a more direct support for the heteroduplex rejection idea. I think these experiments would strengthen the model, if however the authors believe that they would not be helpful, they may explain why and just discuss this point.

Other minor points:

- Nomenclature of the yeast proteins/genes does not always follow standard formatting
- Fig 5D: the authors conclude that the Sgs1 fusions "eliminate D-loop formation". It could be clarified that it may eliminate D-loop formation, but also "disrupt D-loops", which would both reduce the steady-state D-loop levels, correct? The wording used by the authors is suggestive (to me) of Sgs1 function upstream of the joint molecule formation. My understanding is that authors favor the latter scenario, as in heteroduplex rejection Sgs1 acts on the joint molecules.

Referee #2:

In the manuscript by Xie et al, the authors examine contribution of Mec1 signaling to gross chromosomal rearrangements and identify a role for checkpoint signaling in resection control and also through targeting of Sgs1. The authors find that Sgs1 ability to interact with Dbp11, which is mediated by Mec1 is critical for preventing GCRs as is Sgs1 helicase activity. The paper presents some very complicated genetic approaches to analyze how Sgs1 activity prevents GCRs in the absence of Mec1. Overall, the paper presents interesting findings but it is difficult to follow. The genetic background and mutants are constantly changing and the rationales behind the experiments are complex. Perhaps a few reminders of why specific genetic backgrounds are used would make it clearer and the models certainly help. Other specific comments are addressed below.

Comments:

1) It is confusing that the DNA binding domain of Sgs1 (HRDC) is dispensable for GCRs in the *ddc1ΔMAD dna2-aa tel1Δ* strain if the model is that Sgs1 function in heteroduplex rejection is required for its role in suppressing GCRs. Perhaps this could be better explained.

2) The expression of the plethora of Sgs1 mutants used in this study is not tested and may be an important caveat if specific activities are needed for GCR suppression. For example, what is the expression level of Sgs1-9Mut and Sgs1-ΔAP in Figure 2? How does the GCR rate of the *exo1Δ sgs1-9mut* compare to a *exo1Δ sgs1Δ*? Similarly in Figure 4B, do the CDK or Mec1 phosphorylation Sgs1 mutants lead to destabilized protein that is independent of its phosphorylation status?

3) The assay in Figure 5C and what is actually graphed are not well described. As this is a critical part of the model, the assay should be discussed in greater detail. Also some of the mutants that were found to be critical for Sgs1 function could also be analyzed in this assay (such as the helicase dead, etc), this data would help support this part of the model.

4) Why specific color text in Figure 6 are used should be described in the legend.

5) The text is complicated to read and the rational behind some of the mutant backgrounds are hard to follow. Perhaps extending the models in Figure 1F (which were very helpful) to some of the other mutant backgrounds in the other figures that you are testing would help with the flow. For example, the *tel1Δ rad9Δ exo1Δ sgs1Δ*, *ddc1ΔMAD dna2-aa tel1Δ*, *ddc1Δ tel1Δ rad53Δ* are difficult genotypes to understand for a non-expert and then all these mutants are added in. It is confusing for the reader and for a non-checkpoint expert.

Referee #3:

In this manuscript, the authors performed a follow-up study based on their previous works to examine the mechanisms by which Mec1-dependent signaling prevents GCRs in *S. cerevisiae*. The authors exploited their previous phosphoproteomic approaches and genetic experiments showing that upon DNA hyper-resection, Mec1 becomes hyper-activated and regulates the STR complex to control homologous recombination-dependent DNA repair (Sanford et al. EMBO J. 2021, PMID 33764556, ref 54). In this manuscript, the authors provide evidence that Mec1 signaling prevents GCRs by at least two pathways: by activating the checkpoint kinase Rad53 and by controlling the recruitment of Sgs1 to single-stranded DNA lesions via phosphorylation and the

scaffold complex Dbp11-9-1-1. The authors provide elegant experiments to establish that the artificial targeting of Sgs1 to ssDNA bypasses Mec1-signaling to suppress GCRs when the checkpoint kinase Rad53 or the anti-resection factor Rad9 are not functional. Moreover, the authors provide molecular evidence that targeting Sgs1 to ssDNA limits the capture of D-loop event. Combined with the fact that the helicase activity of Sgs1 is required to limit GCRs, the data suggest a model in which Mec1-dependent signaling prevents GCRs by ensuring Sgs1 recruitment to ssDNA lesions to promote heteroduplex rejection. Although this work provides new insights into how the Mec1/ATR kinase maintains genome stability, several major problems hamper the overall coherence of the study.

A first major concern relates to the fact that hyper-resection is supposed to trigger the Mec1-dependent Sgs1 salvage pathway to limit GCRs (models on figure 6). This conclusion appears not well-supported by the data. Even though tel1 rad9 sgs1 or tel1 rad53 sgs1 triple mutant exhibit a high level of GCR rates, this is not related to the inability of Mec1 to phosphorylate Sgs1 or mediate its recruitment to Dbp11 (Figure 1, 2 and S3). Indeed, the lack of Mec1-dependent phosphorylation events on Sgs1 or the lack of Dbp11-Sgs1 interaction is not sufficient to trigger an increase in the rate of GCRs. The critical role of Mec1 signaling in regulating Sgs1 is revealed only in the absence of the nuclease Exo1, a situation in which Mec1-dependent phosphorylation of Sgs1 is reduced (as shown by the authors in Sanford et al. EMBO J. 2021, PMID 33764556, ref 54) because of a reduced amount of ssDNA. It is therefore unjustified that the authors conclude that hyper-resection triggers an Sgs1 salvage pathway to limit GCRs. This reviewer acknowledges the fact that the authors mention "The reason for the increased importance of Sgs1 phosphorylation in the absence of EXO1 remains unclear (page 10)" but it is critical that the authors solve this point. Is the role of Exo1 in GCR prevention in rad9 tel1 or rad53 tel1 related to its nuclease activity or to a structural role?

A second major concern is related to the claim that Mec1 signaling on Sgs1 promotes heteroduplex rejection to limit non-allelic homeologous recombination events. Since the authors did not analyze the structure of GCR events, there is no evidence of this. While the GCR analysis provided in this paper is only quantitative, a structural analysis of the GCRs is crucially missing to support the main message of the paper: Mec1-Sgs1 pathway suppresses GCR formation by promoting heteroduplex rejection.

A third major concern is related to the fact that all the GCR experiments were performed in the absence of the checkpoint kinase Tel1 which plays a critical role in limiting GCRs in the absence of Mec1. Moreover, Tel1 regulates telomere addition during GCR formation, and its deletion (combined with other deletions) may result in other repair defects that will impact overall the GCR outcome (rate and mechanisms of formation). Therefore, how do the authors conclude that the role of Sgs1 in preventing GCRs upon hyper-resection is related to Mec1-signaling without a contribution from Tel1 activity ?

A fourth major concern relates to the interpretation of the results. The data are not thoroughly explained in the result section and in several cases the conclusions emanate from an overinterpretation of the results.

Specific points:

1. The GCR rates are not consistent with previous studies. The authors refer to the PMID: 11239397 paper for the GCR rates in mec1 Δ . The rates in the current paper and PMID: 11239397 and PMID: 29899143 (a paper from the same lab) do not match. For example, the GCR rate in tel1 rad53 mutant is 10 times higher in the two published studies than in the present paper. The GCR rate in the single tel1 mutant and rad9 mutant is not 0. Also, it is not clear why all controls are not shown in the present study (WT, mec1 Δ , ...), as expected for a proper analysis of the data.

2. Figure 1D: It is unclear why the hits of up-regulated Mec1-dependent phosphorylation events common to rad53 and rad9 backgrounds refer to as "response to resection defects". Do the authors mean that these hits reflect a defective resection or an uncontrolled resection? Based on the schemes presented on Figure 1F, these hits should be referred to as "Response to de-regulated resection".

3. Figure 1E: Are rad9 and rad53 epistatic in limiting GCRs ? In other words, do they work in the same pathway for GCR suppression (model in figure 1F)? Why Sgs1 suppresses more efficiently GCRs in rad9 mutant?

4. Figure 1F: To verify the model, a comparison between mec1 Δ and rad53 Δ sgs1 Δ is required.

5. Figure 2 and Page 9: "our findings also suggest that upon loss of DNA end resection control via Rad53 or Rad9, the Mec1-Sgs1 pathway functions as a salvage response important to limit GCRs." Data presented on Figure 2 and S3 are not supporting these conclusions. The lack of Mec1-dependent phosphorylation events on Sgs1 or the lack of Dbp11-Sgs1 interaction is not sufficient to trigger a high rate of GCRs. It does so only when the nuclease Exo1 is deleted. How does the quadruple mutant tel1 sgs1 exo1 rad9 or tel1 sgs1 exo1 rad53 behave in this GCR assay ?

6. Figure 2C: It is concluded that the GCRs originate due to deregulated HR (GCR rate decreased upon deletion of rad52). The genetic background used (tel1 mutant) allows to recover GCRs formed by mechanisms relying on DSB repair and recombination since telomere addition is impaired. It is only logical that the GCRs are dependent on Rad52.

7. Figure S3: The median rate in tel1 rad53 sgs1 + Sgs1 is 0. Why is it drastically different from the GCR rate in tel1 rad53 mutant (Figure 1E).

8. Figure 3: The fusion RBD-Sgs1 rescues the high rate of GCRs of the tel1 rad9 exo1 sgs1 mutant (deregulated resection) and of the ddc1 delta MAD dna2-aa tel1 mutant (Mec1 signaling defect). The interpretation of the authors is that the artificial targeting of Sgs1 to ssDNA lesions bypasses the requirement of Mec1 signaling to prevent GCRs. It would strengthen the conclusion if the authors could show that RBD-Sgs1 is efficiently recruited to ssDNA in vivo. Data provided in figure 3D-F rather indicates that the expression of RBD-Sgs1 leads to the accumulation of DNA damage.

9. Figure 3C: How was BIR measured?

10. Figure 4H-I: The authors conclude that cells lacking Mec1 accumulate GCRs because of an inability of Sgs1 to be properly recruited. The data pointing to this conclusion is rather indirect. What is the GCR rate in mec1 mutant expressing Sgs1-RBD? One would expect that Sgs1-RBD expression leads to a partial suppression of the GCR rates in mec1 mutant.

11. Figure 5A-B: The authors provide data to support that RBD-Sgs1 efficiently suppresses Rad52-dependent GCRs but not Rad51-dependent GCRs, although it is not clear why the authors switch to a ddc1 tel1 rad53 background to test this, a background in which wild type sgs1 is present. It would have been more convincing to use the tel1 rad53 exo1 sgs1 (deregulated resection) or ddc1 delta MAD dna2-aa tel1 (defective Mec1 signaling). Moreover, the GCRs rate in a ddc1 tel1 rad53 mutant is of 7 508 and 100 % of those GCR events are suppressed by expressing RBD-Sgs1 (Figure 4B). This GCR rate drops to 1 857 in ddc1 tel1 rad53 rad51 mutant, indicating that 75 % of the GCRs are Rad51-dependent (Figure 5A, left panel). This GCR rate drops to 1 089 in a ddc1 tel1 rad53 rad52, indicating that 86 % of those GCR events are dependent on Rad52. One interpretation is that most of GCRs events, prevented by Mec1-signaling and by expressing RBD-Sgs1, are Rad51 and Rad52 dependent. Overall, some inconsistency appears when comparing those data and requires clarification from the authors.

12. "our results showed that engineered Sgs1 recruitment suppresses HR driven GCRs through enhanced heteroduplex rejection" (page 14). The data do not explicitly show this. Without a structural analysis of the GCRs, this conclusion cannot be supported. Determining the structure of GCRs is important for understanding the mechanisms by which they are formed.

13. Statistics are missing for all GCR rates.

14. How the authors explained that the GCR rate in several genetic backgrounds correspond to zero. In some cases, this does not match their own previous results. Does it mean that GCRs are not formed at all or that GCR cannot be recovered ?

15. Is SML1 deleted in rad53 strains?

16. Figure S2: The figure is not enough explained in the text.

17. Figure S3A is not mentioned in the text.

18. This reviewer could not get access to table S1 to S3.

General Response to Referees

We thank all the referees for carefully reading our manuscript and highlighting that our findings are interesting and represent an important contribution to the field. The critiques were very useful and have helped us improve the paper. We have addressed all the points brought by the referees in the point-by-point response below. Of importance, we were able to provide additional evidence supporting a key role for Mec1 in regulating the heteroduplex rejection function of Sgs1 (new Figure 6). In addition, as explained below, a range of novel data were provided to address the points raised by the referees.

Our answers to referees' points are indicated below in blue. In the manuscript, we have highlighted in blue all new text added.

Referee #1

1. The authors note that the canonical (in wild type cells) function of Mec1 in the suppression of GCRs involves Rad53 and its function in resection control, and cite relevant literature. The model that the Sgs1 backup pathway functions through heteroduplex rejection is reasonable, but its support is rather indirect (e.g. the function does not involve Top3). Sgs1 also functions in resection together with Dna2. Does the resection role of Sgs1 play any role in this context? The authors could try eliminating Dna2 (or its nuclease activity, with the necessary pif1-m2 suppressor mutation), which would exclude a resection function. On the other hand, testing a genetic interaction of the sgs1 mutation(s) in combination with e.g. msh2-d could provide a more direct support for the heteroduplex rejection idea. I think these experiments would strengthen the model, if however the authors believe that they would not be helpful, they may explain why and just discuss this point.

We thank the reviewer for bringing up these interesting points.

1) The effect of Dna2-nd on GCR suppression.

As suggested by the referee, we disrupted the nuclease activity of Dna2 by introducing the E675A mutation (nuclease dead mutation, *dna2-nd*, PMID: 10748138). As shown in

the dilution assay below, *dna2-nd* cells display a growth defect. Since *DNA2* is an essential gene with key roles in DNA replication (PMID: 7644470), we interpret our result as the E675A mutation affecting resection-independent functions of DNA2, likely in DNA replication. Consistent with this notion, *dna2-nd sgs1Δ* cells show a severe growth defect (see below) and the *dna2-nd* mutation per se already results in cell lethality, unless PIF1 is deleted. Based on these findings, we believe that *dna2-nd* is not a clean separation-of-function mutation to assess the resection role of Sgs1 in GCR suppression and to distinguish between GCRs derived from deregulated resection and GCRs derived from perturbed DNA replication.

2) Genetic interaction between Sgs1^{9mut} and Msh2

We tested the genetic interaction between Sgs1^{9mut} and *msh2Δ* (new Fig. S13). *msh2Δ* does not further increase the GCR rate of Sgs1^{9mut}, supporting our model that Mec1 regulates Sgs1's function in heteroduplex rejection. If Sgs1^{9mut} induces GCRs via resection, an increase in GCR rate should be detected in *msh2Δ* Sgs1^{9mut} cells. In addition, Sgs1 contributes to the rejection of homeologous recombination in *msh2Δ* cells in a manner that depends on the 9 S/T-Q sites mutated in Sgs1^{9mut} (new Fig. S13B-D), further suggesting that Sgs1 has Msh2-independent roles in heteroduplex rejection in the context of GCRs. In agreement with our results, a previous finding from Kolonder's group (PMID: 11138010) reported that Sgs1 and Msh2 exhibited synergistic effects in suppressing both GCRs and homeologous recombination.

We have added text to report and discuss these new results.

Minor points:

1. *Nomenclature of the yeast proteins/genes does not always follow standard formatting.*

We thank Referee 1 for pointing out this problem, for which we apologize. We have carefully checked and revised the nomenclature of proteins / genes in this manuscript. In short, we use plain text with the first letter capitalized for proteins, and italic text with all letters capitalized for genes.

2. *Fig 5D: the authors conclude that the Sgs1 fusions "eliminate D-loop formation". It could be clarified that it may eliminate D-loop formation, but also "disrupt D-loops", which would both reduce the steady-state D-loop levels, correct? The wording used by the authors is suggestive (to me) of Sgs1 function upstream of the joint molecule formation. My understanding is that authors favor the latter scenario, as in heteroduplex rejection Sgs1 acts on the joint molecules.*

We thank referee 1 for pointing out this. We have changed the word "eliminate D-loop formation" to "lower D-loop levels".

Referee #2

1. *It is confusing that the DNA binding domain of Sgs1 (HRDC) is dispensable for GCRs in the *ddc1ΔMAD dna2-aa tel1Δ* strain if the model is that Sgs1 function in heteroduplex rejection is required for its role in suppressing GCRs. Perhaps this could be better explained.*

We agree that we need to explain this better. We only tested the HRDC mutant in the context of the RBD-sgs1 chimera, in which RBD can recruit Sgs1 to DNA. Thus, it's expected that the HRDC region is not critical in the context of RBD fusion. We added new text to the manuscript to clarify this point. It is worth mentioning that the helicase

domain of Sgs1 has also been shown to have DNA binding ability (PMID: 10366502), which may reduce the dependency on HRDC.

*2. The expression of the plethora of Sgs1 mutants used in this study is not tested and may be an important caveat if specific activities are needed for GCR suppression. For example, what is the expression level of Sgs1-9Mut and Sgs1-ΔAP in Figure 2? How does the GCR rate of the *exo1Δ sgs1-9mut* compared to a *exo1Δ sgs1Δ*? Similarly in Figure 4B, do the CDK or Mec1 phosphorylation Sgs1 mutants lead to destabilized protein that is independent of its phosphorylation status?*

We thank Referee 2 for pointing out this concern. First, all Sgs1 mutant-expression plasmids are using the same endogenous *SGS1* promoter. We have included a western blot showing that all Sgs1 mutants (except for Sgs1^{APΔ}) have similar expression levels as wild type Sgs1 (Fig. S4A). Thus, phosphorylation mutations do not interfere with protein stability. For Sgs1^{APΔ}, we do not favor the idea that the increased GCR is caused by the increased protein level, since we have shown that increasing Sgs1 protein level can suppress GCRs (Figure S7A&B).

In addition to western blot, we also performed an IP-MS experiment comparing the expression of Sgs1 with that of Sgs1^{15mut}. Consistent with the WB, there is no difference in expression between wild type Sgs1 and Sgs1^{15mut}.

3. The assay in Figure 5C and what is actually graphed are not well described. As this is a critical part of the model, the assay should be discussed in greater detail. Also some of the mutants that were found to be critical for Sgs1 function could also be analyzed in this assay (such as the helicase dead, etc), this data would help support this part of the model.

We apologize for skipping the details of the D-loop assay. We have now provided a detailed description of the assay in the text. Concerning other mutants, the effect of Sgs1^{hd} on the DLC signal has been published elsewhere (PMID: 30737186). We now include the DLC results of Sgs1^{9mut} in Fig. 6F.

4. Why specific color text in Figure 6 are used should be described in the legend.

We have made a new model figure eliminating most colors but still highlighting the action of Mec1 (Fig. 7).

5. The text is complicated to read and the rationale behind some of the mutant backgrounds are hard to follow. Perhaps extending the models in Figure 1F (which were very helpful) to some of the other mutant backgrounds in the other figures that you are testing would help with the flow. For example, the tel1Δ rad9Δ exo1Δ sgs1Δ, ddc1ΔMAD

dna2-aa tel1Δ, ddc1Δ tel1Δ rad53Δ are difficult genotypes to understand for a non-expert and then all these mutants are added in. It is confusing for the reader and for a non-checkpoint expert.

We apologize for not explaining the different genotypes used in this manuscript well. We have revised the text in the manuscript and added more explanations to explain the genotypes and underlying rationale.

Referee #3

1. A first major concern relates to the fact that hyper-resection is supposed to trigger the Mec1-dependent Sgs1 salvage pathway to limit GCRs (models on figure 6). This conclusion appears not well-supported by the data. Even though tel1 rad9 sgs1 or tel1 rad53 sgs1 triple mutant exhibit a high level of GCR rates, this is not related to the inability of Mec1 to phosphorylate Sgs1 or mediate its recruitment to Dbp11 (Figure 1, 2 and S3). Indeed, the lack of Mec1-dependent phosphorylation events on Sgs1 or the lack of Dbp11-Sgs1 interaction is not sufficient to trigger an increase in the rate of GCRs. The critical role of Mec1 signaling in regulating Sgs1 is revealed only in the absence of the nuclease Exo1, a situation in which Mec1-dependent phosphorylation of Sgs1 is reduced (as shown by the authors in Sanford et al. EMBO J. 2021, PMID 33764556, ref 54) because of a reduced amount of ssDNA. It is therefore unjustified that the authors conclude that hyper-resection triggers an Sgs1 salvage pathway to limit GCRs. This reviewer acknowledges the fact that the authors mention " The reason for the increased importance of Sgs1 phosphorylation in the absence of EXO1 remains unclear (page 10)" but it is critical that the authors solve this point. Is the role of Exo1 in GCR prevention in rad9 tel1 or rad53 tel1 related to its nuclease activity or to a structural role?

We thank the reviewer for raising these points. It is important that we clarify them.

First, the referee mentions “Indeed, the lack of Mec1-dependent phosphorylation events on Sgs1 or the lack of Dbp11-Sgs1 interaction is not sufficient to trigger an increase in the rate of GCRs. The critical role of Mec1 signaling in regulating Sgs1 is revealed only in the absence of the nuclease Exo1”. It’s true that Sgs1 mutants display a strong increase in GCR rates in the absence of Exo1. However, when Rad9 is further removed, the rate is significantly increased (Fig. 2), supporting our model that the Mec1-Sgs1 salvage pathway becomes important upon deregulated resection.

Second, the referee notes “The critical role of Mec1 signaling in regulating Sgs1 is revealed only in the absence of the nuclease Exo1, a situation in which Mec1-dependent phosphorylation of Sgs1 is reduced (as shown by the authors in Sanford et al. EMBO J. 2021, PMID 33764556, ref 54) because of a reduced amount of ssDNA”. In fact, Sanford et al shows that Mec1-dependent phosphorylation of Sgs1 in *rad9Δ* cells decreases only when **both** Exo1 and Dna2 are defective. Moreover, a previous report (PMID: 25637499) showed that resection in *rad9Δ* cells depend mainly on Sgs1 but not Exo1, further consistent with the idea that deletion of *EXO1* in *rad9Δ* cells does not substantially affect ssDNA exposure and Sgs1 phosphorylation by Mec1.

Following the reviewer’s suggestion, we have assessed the role of the exonuclease function of Exo1 by measuring GCR rates of cells carrying a nuclease dead mutation allele of *EXO1* (*D173A*). As shown in the new Fig. S5, the *D173A* mutation dramatically induces GCRs in the absence of Sgs1, though not as strong as *exo1Δ*, indicating that the nuclease activity (potentially through resection) is critical for inhibiting most GCR events. However, the detailed mechanisms of Exo1-mediated GCR suppression are still unclear and require further investigation.

We have added text to mention and better explain these points.

2. A second major concern is related to the claim that Mec1 signaling on Sgs1 promotes heteroduplex rejection to limit non-allelic homeologous recombination events. Since the authors did not analyze the structure of GCR events, there is no evidence of this. While

the GCR analysis provided in this paper is only quantitative, a structural analysis of the GCRs is cruelly missing to support the main message of the paper: Mec1-Sgs1 pathway suppresses GCR formation by promoting heteroduplex rejection.

This is a fair point. To directly test our model that Mec1 signaling affects the ability of Sgs1 to reject homeologous recombination events, we performed a single-strand annealing assay (new Fig. 6A-D). Our results show that mutation of the 9 S/T-Q sites in Sgs1 reduces the ability of Sgs1 to reject heteroduplexes formed between homeologous sequences. In addition, we have shown that RBD-Sgs1 suppresses GCR via disrupting D-loop formation (Fig. 5D & Fig. S12A), and that RBD-Sgs1^{9mut} impairs the GCR suppression (Fig. 4B). Moreover, we have also included new data testing epistasis between Sgs1^{9mut} and *msh2Δ* (new Fig. S13). We find that *msh2Δ* does not further increase the GCR rate of Sgs1^{9mut}, supporting the model that Mec1 regulates Sgs1's function in heteroduplex rejection. If Sgs1^{9mut} induces GCRs via resection, an increase in GCR rate should be detected in *msh2Δ* Sgs1^{9mut} cells.

We acknowledge that analyzing GCR structure would be informative. However, sequencing survivors and analyzing the GCR structure are not simple and would require extensive additional work that is beyond our current expertise.

Overall, in light of these new results we added to the revised manuscript, we believe it's reasonable to propose that Mec1 phosphorylates Sgs1 to promote the rejection of homeologous recombination, thereby suppressing GCRs.

We have added text to report and discuss these new results.

3. A third major concern is related to the fact that all the GCR experiments were performed in the absence of the checkpoint kinase Tel1 which plays a critical role in limiting GCRs in the absence of Mec1. Moreover, Tel1 regulates telomere addition during GCR formation, and its deletion (combined with other deletions) may result in other repair defects that will impact overall the GCR outcome (rate and mechanisms of

formation). Therefore, how do the authors conclude that the role of Sgs1 in preventing GCRs upon hyper-resection is related to Mec1-signaling without a contribution from Tel1 activity?

We agree that understanding the contribution of Tel1 to GCR suppression is a very interesting direction for future work. We have opted to focus on Mec1 for this work given its more important role compared to Tel1 and because of the following reasons: a) Deletion of Tel1 alone does not increase GCRs whereas deletion of Mec1 alone does, indicating that Mec1 plays a more important and primary role in GCR suppression; b) Sgs1 is phosphorylated by Mec1, not by Tel1 (PMID 33764556; PMID 25752575) and; c) *rad53Δ* does not induce an up-regulation of Tel1 signaling (unpublished data in the lab). Thus, current evidence is consistent with the view that Tel1 does not regulate Sgs1 to suppress GCRs. Nonetheless, we acknowledge that Tel1 plays important roles in preventing GCRs, especially in cells lacking Mec1. We have mentioned Tel1 in the discussion: “Moreover, it will be important to define the role of Tel1 in limiting GCR accumulation upon loss of Mec1. One possibility is that DSBs accumulate in *mec1Δ* cells due to increased fork collapse, and that Tel1 is required to properly repair these breaks and prevent them from engaging in deleterious DNA transactions that cause GCRs.”

4. A fourth major concern relates to the interpretation of the results. The data are not thoroughly explained in the result section and in several cases the conclusions emanate from an overinterpretation of the results.

We apologize for the lack of clarity. We have gone over the text thoroughly and carefully revised it to tone down some claims and prevent overinterpretations.

Specific points:

1. The GCR rates are not consistent with previous studies. The authors refer to the PMID: 11239397 paper for the GCR rates in *mec1Δ*. The rates in the current paper and

PMID: 11239397 and PMID: 29899143 (a paper from the same lab) do not match. For example, the GCR rate in tel1 rad53 mutant is 10 times higher in the two published studies than in the present paper. The GCR rate in the single tel1 mutant and rad9 mutant is not 0. Also, it is not clear why all controls are not shown in the present study (WT, mec1Δ, ...), as expected for a proper analysis of the data.

We thank Referee 3 for raising this point that requires our clarification. First, Lanz et al (PMID: 29899143) does not measure the GCR rate of *tel1Δ rad53Δ*. They directly used the rate in the published paper (PMID: 11239397). As for the difference in rates between our manuscript and the initial paper (PMID: 11239397), one explanation is that we use a modified GCR assay (see methods). We plate 50~100 fold fewer cells than the original protocol. Although this modified assay allows us to test more independent isolates without using lots of GCR plates, it is not ideal for measuring low GCR rates (especially $<500 \times 10^{-10}$) given the stochasticity of survivor colonies appearing. We have added text to the legend of figure 1 explaining this caveat of our modified GCR assay. We emphasize that this caveat does not affect our analyses since all strains relevant for our work display GCR rates above 500×10^{-10} .

For the “0” rate, we apologize for the confusion. “0” meant zero detected GCR events in our assay. As explained above, since we use fewer cells in the assay, we do have a lower probability of detecting GCR events and are unable to accurately evaluate strains with low GCR rates. We have changed “0” to “not detected”.

As requested by the reviewer, we now show the GCR rate of *mec1Δ* cells in the new Fig. S3. We decided to not show it in the main figure since the main message of Figure 1E is to show that Rad53/Rad9 and Sgs1 work in parallel to suppress GCRs.

2. Figure 1D: It is unclear why the hits of up-regulated Mec1-dependent phosphorylation events common to rad53 and rad9 backgrounds refer to as "response to resection

defects". Do the authors mean that these hits reflect a defective resection or an uncontrolled resection? Based on the schemes presented on Figure 1F, these hits should be referred to as "Response to de-regulated resection".

We apologize for this wording issue. We have changed "response to resection defects" to "response to deregulated resection" in Figure 1D.

3. Figure 1E: Are rad9 and rad53 epistatic in limiting GCRs? In other words, do they work in the same pathway for GCR suppression (model in figure 1F)? Why Sgs1 suppresses more efficiently GCRs in rad9 mutant?

We thank referee 3 for noting this point. We have measured the GCR rate of *rad9Δ rad53Δ sgs1Δ* cells and added this new result to Fig. 1E. The result indicates that Rad9 and Rad53 are epistatic in limiting GCRs.

The reason why Sgs1 suppresses GCRs more efficiently in *rad9Δ* cells could be that *rad9Δ* cells generate more ssDNA than *rad53Δ* cells so Mec1 signaling gets more strongly activated, or that *rad53Δ* cells accumulate residual GCRs that are caused by fork collapse.

4. Figure 1F: To verify the model, a comparison between mec1Δ and rad53Δ sgs1Δ is required.

We thank Referee 3 for suggesting this experiment. We have performed this experiment (new Fig. S3). *mec1Δ* shows higher GCR rate than *rad53Δ sgs1Δ*, suggesting that Mec1 regulates additional substrates to suppress GCRs. We have mentioned "other substrates" in the legend of Figure 7.

5. Figure 2 and Page 9: "our findings also suggest that upon loss of DNA end resection control via Rad53 or Rad9, the Mec1-Sgs1 pathway functions as a salvage response

important to limit GCRs." Data presented on Figure 2 and S3 are not supporting these conclusions. The lack of Mec1-dependent phosphorylation events on Sgs1 or the lack of Dbp11-Sgs1 interaction is not sufficient to trigger a high rate of GCRs. It does so only when the nuclease Exo1 is deleted. How does the quadruple mutant tel1 sgs1 exo1 rad9 or tel1 sgs1 exo1 rad53 behave in this GCR assay?

We acknowledge that the requirement of Mec1-mediated Sgs1 regulation in GCR suppression becomes more pronounced in *exo1Δ* cells. However, when Rad9 is further removed, the phenotype becomes significantly more striking, consistent with our model that the Mec1-Sgs1 salvage pathway becomes important upon deregulated resection.

Based on our measurements, the median GCR rate of *tel1Δ sgs1Δ exo1Δ* is ~80,000. The GCR rate increases to over 110,000 when *RAD9* is further deleted (new Fig. S5).

6. Figure 2C: It is concluded that the GCRs originate due to deregulated HR (GCR rate decreased upon deletion of rad52). The genetic background used (tel1 mutant) allows to recover GCRs formed by mechanisms relying on DSB repair and recombination since telomere addition is impaired. It is only logical that the GCRs are dependent on Rad52.

We believe that under the condition of deregulated resection, most GCRs are induced by non-allelic HR, as Figure 2C shows that *rad52Δ* removes most GCRs in our strain. But we still acknowledge the possibility that a small number of GCR events can still be driven by NHEJ (PMID: 28684602) or telomere addition (telomere addition can still happen in *tel1Δ rad9Δ* cells (PMID: 11917116)). We therefore believe that the data showing decreased GCR upon *RAD52* deletion is quite relevant.

7. Figure S3: The median rate in tel1 rad53 sgs1 + Sgs1 is 0. Why is it drastically different from the GCR rate in tel1 rad53 mutant (Figure 1E).

We thank referee 3 for bringing this concern. In Figure 1E we grow cells in YPD while in Figure S3 (now new Figure S4) we use drop-out media for plasmid expression. Under

the same OD600, YPD has much higher cell density than drop-out media. Since we plate cells based on OD, less cells will be plated when using plasmid, making it harder to recover a GCR event if the strain has relatively low GCR rates when using drop-out media. And again, “0” means “not detected”.

8. Figure 3: The fusion RBD-Sgs1 rescues the high rate of GCRs of the tel1 rad9 exo1 sgs1 mutant (deregulated resection) and of the ddc1 delta MAD dna2-aa tel1 mutant (Mec1 signaling defect). The interpretation of the authors is that the artificial targeting of Sgs1 to ssDNA lesions bypasses the requirement of Mec1 signaling to prevent GCRs. It would strengthen the conclusion if the authors could show that RBD-Sgs1 is efficiently recruited to ssDNA in vivo. Data provided in figure 3D-F rather indicates that the expression of RBD-Sgs1 leads to the accumulation of DNA damage.

Referee 3 brings a relevant point. Previous study (PMID: 29033322) has shown the structural and biochemical basis of RBD-RPA interaction. Thus, we believe that RBD fusion should enable recruiting Sgs1 to ssDNA. Furthermore, our data (Fig. 3F&G) shows that RBD-Sgs1, but not Sgs1 alone, induces elevated Rad52 foci. This increased Rad52 foci is not caused by increased DSBs, as Rad53 is not activated (Fig. 3H). Together with the fact that RBD-Sgs1 can disrupt D-loop (Fig. 5D & Fig. S12A), it's reasonable to propose that RBD recruits Sgs1 to ssDNA, disrupting D-loop and leading to persistent Rad52 foci.

9. Figure 3C: How was BIR measured?

We thank referee 3 for pointing this out, for which we apologize. An illustration of BIR assay used in this study has been added (Fig. 3C).

10. Figure 4H-I: The authors conclude that cells lacking Mec1 accumulate GCRs because of an inability of Sgs1 to be properly recruited. The data pointing to this conclusion is rather indirect. What is the GCR rate in mec1 mutant expressing Sgs1-

RBD? One would expect that Sgs1-RBD expression leads to a partial suppression of the GCR rates in mec1 mutant.

Referee 3 brings a relevant point. We have expressed Sgs1 and RBD-Sgs1 in *mec1Δ* cells. However, as the base GCR rate of *mec1Δ* is not very high, we failed to recover GCRs in both groups. To increase the basal GCR rate for detection, we generated *mec1Δ sgs1Δ* strains, where we expressed Sgs1^{APΔ} and RBD-Sgs1^{APΔ} to compare. As shown below, the RBD fusion can suppress GCRs in *mec1Δ* cells. We note that we were not able to examine GCRs rates in cells lacking both *MEC1* and *TEL1* since plasmid transformation resulted in massive loss of cell viability.

11. *Figure 5A-B: The authors provide data to support that RBD-Sgs1 efficiently suppresses Rad52-dependent GCRs but not Rad51-dependent GCRs, although it is not clear why the authors switch to a ddc1 tel1 rad53 background to test this, a background in which wild type sgs1 is present. It would have been more convincing to use the tel1 rad53 exo1 sgs1 (de-regulated resection) or ddc1 delta MAD dna2-aa tel1 (defective Mec1 signaling). Moreover, the GCRs rate in a ddc1 tel1 rad53 mutant is of 7 508 and 100 % of those GCR events are suppressed by expressing RBD-Sgs1 (Figure 4B). This GCR rate drops to 1 857 in ddc1 tel1 rad53 rad51 mutant, indicating that 75 % of the GCRs are Rad51-dependent (Figure 5A, left panel). This GCR rate drops to 1 089 in a ddc1 tel1 rad53 rad52, indicating that 86 % of those GCR events are dependent on Rad52. One interpretation is that most of GCRs events, prevented by Mec1-signaling and by expressing RBD-Sgs1, are Rad51 and Rad52 dependent. Overall, some inconsistency appears when comparing those data and requires clarification from the authors.*

We thank referee 3 for mentioning this point. We have done this experiment using *tel1Δ rad9Δ exo1Δ sgs1Δ rad51Δ/52Δ* (Fig. S11). As *tel1Δ rad9Δ exo1Δ sgs1Δ* + Sgs1 has very low GCR rate (median is 596×10^{-10} , Fig. 2), further deletion of *RAD51* or *RAD52* decreases the GCR rate to undetectable level. Thus, we decided to use Sgs1^{APΔ}. As shown in Figure S11, RBD-Sgs1^{APΔ} shows detectable enhancement in GCR suppression compared with Sgs1^{APΔ} in *rad51Δ* cells. However, there is no difference between RBD-Sgs1^{APΔ} and Sgs1^{APΔ} in *rad52Δ* cells, consistent with data in Figure 5A-B.

As for the numbers, in *ddc1 tel1 rad53* cells, Rad51 accounts for 75% of GCRs while Rad52 accounts for 86% of GCRs. We believe these numbers are reasonable and consistent. First, these numbers confirm that most GCRs in *ddc1 tel1 rad53* cells are driven by HR. Second, Rad52-dependent HR pathways include Rad51-dependent HR and single-strand annealing (SSA). Thus, it's reasonable that Rad52 accounts for a higher proportion than Rad51.

12. "our results showed that engineered Sgs1 recruitment suppresses HR driven GCRs through enhanced heteroduplex rejection" (page 14). The data do not explicitly show this. Without a structural analysis of the GCRs, this conclusion cannot be supported. Determining the structure of GCRs is important for understanding the mechanisms by which they are formed.

We thank referee 3 for suggesting this experiment. We have addressed this point in the major concern #2 above. We have also toned down our claim to say that "our results suggest".

13. Statistics are missing for all GCR rates.

We thank referee 3 for bringing this point, for which we apologize. We have added statistics in all figures, except for Figure 4A & C. In Figure 4A & C, we believe that the

numbers are sufficient to show the difference. Adding statistics will generate many lines and make the graph very busy.

Since we mainly compare 2 groups (e.g., Sgs1^{wt} vs Sgs1^{mut}), we use two-tailed t-test for statistics.

14. How the authors explained that the GCR rate in several genetic backgrounds correspond to zero. In some cases, this does not match their own previous results. Does it mean that GCRs are not formed at all or that GCR cannot be recovered?

We have addressed this point in the specific point #1 above. In short, “0” means no detected GCR events in our assay. We have changed all “0” to “not detected”.

15. Is SML1 deleted in rad53 strains?

All the GCR strains used in this study have *sm1*Δ.

16. Figure S2: The figure is not enough explained in the text.

We thank referee 3 for bringing this point, for which we apologize. We have explained Figure S2 in greater details.

17. Figure S3A is not mentioned in the text.

We thank referee 3 for bringing this point, for which we apologize. We have included an explanation for Figure S3A (now Fig. S4B).

18. This reviewer could not get access to table S1 to S3.

We apologize for this. Now all supplemental tables should be available.

Dr. Marcus B Smolka
Cornell University
Weill Institute for Cell and Molecular Biology
339 Weill Hall
Ithaca, NY 14853-7202

17th May 2024

Re: EMBOJ-2023-116302R
Multi-Step Control of Homologous Recombination Via Mec1/ATR Suppresses Chromosomal Rearrangements

Dear Marcus,

Thank you for submitting a revised version of your manuscript. All three referees (see below) are fully satisfied with your responses and revisions, and we shall therefore be happy to accept your manuscript for EMBO Journal publication, as soon as the following, remaining editorial issues have been addressed:

- Please upload all main Figures and all Expanded View figures as individual files, with sufficient resolution/quality for production.
- Please adjust the format of the reference list and of the in-text citations according to EMBO Journal format (alphabetical order, author name et al + year...)
- On the abstract page of the manuscript, please include 4-5 general keyword terms to enhance searchability.
- Please rename the Conflict of Interest section into "Disclosure and Competing Interests Statement", in accordance with our updated Guide to Authors (<https://www.embopress.org/competing-interests/>)
- As we are switching from a free-text author contribution statement towards a more formal statement based on Contributor Role Taxonomy (CRediT) terms, please remove the present Author Contribution section and instead specify each author's contribution(s) directly in the Author Information page of our submission system during upload of the final manuscript. See <https://casrai.org/credit/> for more information.
- Please make sure to at this stage remove reviewer access information for the PRIDE deposition from the Data Availability section, and ensure that the data become openly available latest at the time of online publication.
- Please double-check to make sure to all relevant funding information in the manuscript is congruent with the info entered into our submission system (fellowship from the Academia Sinica may need to be entered as a funder in eJP).
- For the 3 EV Datasets uploaded as XLSX files, each should contain a separate "Legends" tab mentioning its name and explaining the contents. Furthermore, while Dataset EV1 clearly is a dataset, I feel that the other two would be more appropriately converted to Expanded View Tables - i.e. renamed to Table EV1/2 both within the respective files and at all in-text callouts.
- In the Source Data checklist, please use the free-text box to clarify where relevant Source Data (e.g. for Figures 1B & 1D), which has not been included in the Source Data uploads, can be found (PRIDE with accession code, or EV Dataset, etc...).
- Finally, during routine pre-acceptance checks, our data editors have raised the following queries regarding figures, data, and legends, which I would ask you to address (ideally using the Track Changes option):
 - * Please define the annotated p values ****/**/*/* in the legend of figure 2b-d; 3d, g, i-j; 4b; 5a; 6d-f; as appropriate.
 - * Please note that n=2 in figure 5d -> therefore, no statistical tests can be applied and error bars have to be removed, instead, individual data points need to be plotted
 - * Please note that scale bar and its definition are missing for figure 3f.

I am therefore returning the manuscript to you for a final round of revision, to allow you to make these modifications and upload the revised files. It would be great if you could resubmit all this by early next week, to ensure a swift further proceeding with formal acceptance and production of the manuscript.

With kind regards,

Hartmut

*** PLEASE NOTE: All revised manuscripts are subject to initial checks for completeness and adherence to our formatting guidelines. Revisions may be returned to the authors and delayed in their editorial re-evaluation if they fail to comply to the following requirements (see also our Guide to Authors for further information):

9) Digital image enhancement is acceptable practice, as long as it accurately represents the original data and conforms to community standards. If a figure has been subjected to significant electronic manipulation, this must be clearly noted in the figure legend and/or the 'Materials and Methods' section. The editors reserve the right to request original versions of figures and the original images that were used to assemble the figure. Finally, we generally encourage uploading of numerical as well as gel/blot image source data; for details see: embopress.org/page/journal/14602075/authorguide#sourcedata

At EMBO Press, we ask authors to provide source data for the main manuscript figures. Our source data coordinator will contact you to discuss which figure panels we would need source data for and will also provide you with helpful tips on how to upload and organize the files.

In the interest of ensuring the conceptual advance provided by the work, we recommend submitting a revision within 3 months (15th Aug 2024). Please discuss the revision progress ahead of this time with the editor if you require more time to complete the revisions. Use the link below to submit your revision:

Link Not Available

Referee #1:

I am happy to recommend the revised manuscript for publication

Referee #2:

The reviewers have now addressed my concerns in the revised manuscript.

Referee #3:

The authors' response to the various criticisms is convincing. They have done an excellent job.

Dr. Marcus B Smolka
Cornell University
Weill Institute for Cell and Molecular Biology
339 Weill Hall
Ithaca, NY 14853-7202

22nd May 2024

Re: EMBOJ-2023-116302R1
Multi-Step Control of Homologous Recombination Via Mec1/ATR Suppresses Chromosomal Rearrangements

Dear Marcus,

Thank you for submitting your final revised manuscript for our consideration. I am pleased to inform you that we have now accepted it for publication in The EMBO Journal!

With kind regards,

Hartmut
